# Effects of *Alnus japonica* Pilot Scale Hot Water Extracts on a Model of Dexamethasone-Induced Muscle Loss and Muscle Atrophy in C57BL/6 Mice

**DOI:** 10.3390/ijms26083656

**Published:** 2025-04-12

**Authors:** Hyeon Du Jang, Chan Ho Lee, Ye Eun Kwon, Tae Hee Kim, Eun Ji Kim, Jae In Jung, Sang Il Min, Eun Ju Cheong, Tae Young Jang, Hee Kyu Kim, Sun Eun Choi

**Affiliations:** 1Department of Forest Biomaterials Engineering, Kangwon National University, Chuncheon-si 24341, Republic of Korea; wkdgusen98@naver.com (H.D.J.); lgh4107@gmail.com (C.H.L.); 2Dr.Oregonin Inc., #802 Bodeum Hall, Kangwondaehakgil 1, Chuncheon-si 24341, Republic of Korea; kye0519@naver.com (Y.E.K.); kth02120@naver.com (T.H.K.); 3Industry Coupled Cooperation Center for Bio Healthcare Materials, Hallym University, Chuncheon-si 24252, Republic of Korea; myej4@hallym.ac.kr (E.J.K.); jungahoo@hallym.ac.kr (J.I.J.); 4Division of Transplantation and Vascular Surgery, Department of Surgery, Seoul National University Hospital, Seoul 03080, Republic of Korea; surgeonmsi@gmail.com; 5College of Forest and Environmental Science, Kangwon National University, Chuncheon-si 24341, Republic of Korea; ejcheong@kangwon.ac.kr (E.J.C.); jty950@naver.com (T.Y.J.); 6Gangwon State Forest Science Institute, 24, Hwamokwon-gil, Chuncheon-si 24207, Republic of Korea; dearkyu@korea.kr

**Keywords:** *Alnus japonica*, natural product, hirsutanonol, hirsutenone, hot water extract, sarcopenia, muscle atrophy

## Abstract

This study investigates the effects of pilot scale Alnus japonica hot water extract (AJHW) on muscle loss and muscle atrophy. Building on previous in vitro studies, in vivo experiments were conducted to evaluate muscle strength, mass, fiber size, protein synthesis, and antioxidant activity. The results showed that AJHW significantly restored muscle strength, increased muscle mass, enhanced the expression of muscle synthesis markers, such as Akt and mTOR, and apoptosis inhibition markers, such as Bcl-2, compared to the muscle atrophy control. Muscle degradation markers, such as Atrogin1, MuRF1, FoxO3α, and the apoptosis activation marker Bax, were decreased compared to the muscle atrophy control. Additionally, AJHW significantly boosted the activity of antioxidant factors like SOD, catalase, and Gpx, suggesting its protective role against oxidative stress-induced muscle damage. The enhanced effects were attributed to the high content of hirsutanonol and hirsutenone, which synergized with oregonin, compounds, identified through phytochemical analysis. While these findings support the potential of AJHW as a candidate for preventing muscle loss, further studies are needed to confirm its efficacy across diverse atrophy models and to elucidate its exact mechanisms.

## 1. Introduction

With the advancement of modern technology, the average life expectancy has increased [1]. Consequently, the proportion of older adults is rising, making health in old age a critical issue. Thus, extensive research on geriatric and chronic diseases related to health in old age is underway. In the context of global aging coupled with a declining birth rate, particularly in the Republic of Korea, the significance of health in old age is escalating [2]. Active research is ongoing on muscles, which play a vital role in body composition, athletic performance, and overall health [3]. Sarcopenia, a condition characterized by decreased muscle synthesis and increased muscle degradation, leads to reduced muscle mass and function [4]. The detrimental effects of sarcopenia include reduced exercise capacity; higher fall accident rates [5]; and the impairment of muscles involved in internal metabolism which increases the risk of metabolic diseases, such as obesity and diabetes, which also adversely affects the heart and cardiovascular system, vital for life’s activities, and which has been reported to elevate the risk of cognitive impairment, cognitive decline, and Alzheimer’s disease [6,7]. In addition, the severity of sarcopenia is increasing, with consistent reports linking it to cancer and other diseases [8,9,10,11,12]. Sarcopenia, linked to various conditions, may result from factors such as nutritional status, underlying medical conditions, exercise habits, and age-related changes [13]. Sarcopenia results from a combination of apoptosis in muscle cells and proteins [14], reduced synthesis of muscle proteins due to reactive oxygen species (ROS) [15], and enhanced degradation of muscle proteins [16]. Research has demonstrated that the signaling pathway system for muscle protein synthesis and degradation involves various factors, including Akt, Atrogin1, MuRF1, FoxO, Myogenin, mTOR, among others, and it is associated with the ubiquitin-proteasome signaling system, apoptosis in muscle proteins, and the caspase pathway [17,18,19].

The treatment for this complex and multifactorial condition includes medication, exercise therapy, and protein intake [20]. However, excessive protein intake and certain medications have been associated with long-term risks, including gout [21] and liver damage [22]. Additionally, anabolic androgenic steroids (AAS), used to treat sarcopenia, pose various adverse effects, such as liver damage [23,24,25], rendering them unsuitable for long-term use [26]. Significantly, the incidence of liver or kidney damage due to excessive protein intake has risen with the popularity of high-protein diets among the elderly and fitness enthusiasts. Furthermore, the misuse or abuse of steroidal drugs for muscle growth has led to severe side effects, including liver and heart damage [27,28,29,30]. Therefore, research is ongoing to develop a sarcopenia treatment that minimizes side effects; however, to date, there have been no successful cases of developing a treatment for sarcopenia with minimal side effects. Therefore, the successful development of new drugs and functional foods for sarcopenia is of great academic and economic value, and various organizations and companies are engaged in this research [31]. One promising research area is natural product pharmaceuticals, which leverage plant secondary metabolites and various plant-derived compounds [32]. These secondary metabolites, chemical defenses produced by plants in response to environmental stresses or natural predators, offer protective benefits [33]. These secondary metabolites contain a variety of components with distinct biological activities in the human body; hence, they are used in developing natural medicines and functional foods [34]. Moreover, plant-derived components have shown sarcopenia-related biological activities, indicating significant potential for developing plant-based sarcopenia treatments [35,36]. Furthermore, various success stories of natural product medicines demonstrate this potential, and it is anticipated that natural product-based therapeutics will increasingly be valued as safe and effective treatments [37,38].

We review several potential natural products for the development of natural product-based therapeutics. Alder from the *Alnus* genus, known for its high biological activity, serves as a material for natural product drug development. The scientific name of alder is “*Alnus Japonica*”, and in Korean herbal medicine, it is used as “Jeog-yang (**赤楊**)” to regulate emotions, stop bleeding, and treat enteritis and diarrhea [39]. Additionally, the *Alnus* genus plant is known internationally as “Red Alder” [40]. Moreover, plants from the *Alnus* genus, including *Alnus japonica*, contain ‘Oregonin’ as an active and indicative compound [41]. Oregonin, a diarylheptanoid compound, is recognized for its antioxidant properties [42]. Plant-derived components, including oregonin, exhibit anti-inflammatory, antitumor, antibacterial, and hepatoprotective effects [43], in addition to antioxidant activities [44,45,46].

In addition, a recently published study on *Alnus japonica*-related muscle loss and muscle atrophy reported that *Alnus japonica* 50% ethanol extract and its indicator component, oregonin, produced by spray-drying 100 kg of *Alnus japonica* bark with 1000 L of 50% EtOH solvent and performing repeated extractions at 80 ± 5 °C for 7 h twice, exhibited activities that promote muscle synthesis and inhibit muscle degradation for the prevention and relief of muscle loss and muscle atrophy [47]. These findings suggest the potential for using *Alnus Japonica*-derived ingredients as functional components for muscle atrophy, alongside verifying their anti-muscle atrophy effects. Accordingly, in our recent study, we conducted a pilot-scale material standardization using domestic tree resources and an investigation into muscle loss and muscle atrophy. We performed biological activity studies focused on oxidative stress, apoptosis, and muscle synthesis and degradation, which influence muscle loss, and carried out preclinical studies including in vivo activity studies, aiming to apply our experience with muscle atrophy-related studies to other raw materials [48,49].

Therefore, this research team utilized 300 kg of *Alnus japonica*, a natural tree resource, and 3000 L of D.W. to perform hot water extraction at a temperature of 100 ± 5 °C for 4 h. The extract was then retrieved by freeze-drying with 10% dextrin to establish a pilot-scale raw material production process. This process aimed to evaluate the efficacy of *Alnus japonica* hot water extract (AJHW) in treating muscle atrophy, develop sarcopenia-specific raw materials, and secure global market competitiveness. Notably, hot water extraction offers advantages over the 50% EtOH extraction method, including enhanced human safety, potential future approval by the Ministry of Food and Drug Safety [50], and increased safety due to the non-use of volatile organic solvents. It also reduces costs by eliminating the need for additional wastewater treatment equipment and staffing for fire or ventilation systems to prevent suffocation accidents. Additionally, the efficacy and activity of extracts vary with the extraction method, necessitating verification of each extract’s efficacy, activity, and content. In this study, we performed phytochemical analysis using HPLC and TLC to monitor oregonin, a major indicator substance of the *Alnus* genus, using Alnus japonica 50% EtOH extract and AJHW extracted on a pilot scale in an Alnus japonica-related muscle loss and muscle atrophy in vitro study [51,52,53,54,55]. The results indicated that AJHW had an equivalent or higher oregonin content compared to the 50% EtOH extract. Conducting a phytochemical analysis under the same conditions as a previous study, we compared the contents of “hirsutanonol” and “hirsutenone”, the aglycone components of oregonin. Oregonin is a bioactive compound known to be derived from plants of the *Alnus* genus. Among the two aglycone forms originating from oregonin, hirsutanonol is formed through the hydrolysis process, where the xylose moiety is removed from oregonin [41,44,51,56,57]. Meanwhile, hirsutenone is generated from hirsutanonol through a dehydration reaction [41,44,51,56,57]. Previous in vitro studies have primarily focused on oregonin, but future research will explore the effects of hirsutenone and hirsutanonol in relation to muscle atrophy, contributing to a more comprehensive understanding of *Alnus* extracts. Therefore, this study requires a phytochemical analysis of hirsutanonol and hirsutenone alongside oregonin.

The findings revealed that AJHW had a higher aglycone content than the 50% EtOH extract, confirming the superiority of the hot water extraction method for the three active compounds: oregonin, hirsutanonol, and hirsutenone. Additionally, we conclude that the hot water extraction method offers significant economic value by reducing production costs and enhancing productivity through a shortened process [58]. We aimed to apply a hot water extraction method using *Alnus japonica*, which possesses high biological activity, to conduct in vivo studies on muscle loss and muscle atrophy, ensuring a stable raw material supply and economic efficiency. Additionally, our previous in vitro study on the biological activity of the *Alnus japonica* hot water extract (AJHW) on dexamethasone induced muscle loss and muscle atrophy was conducted in a controlled environment using cultured cells and blocking external factors, whereas in vivo studies are deemed crucial in preclinical research as they facilitate the identification of more complex biological interactions and long-term environmental influences using animal models [59].

Additionally, to further investigate the preclinical studies on muscle atrophy using compounds previously reported from *Alnus japonica,* dexamethasone-treated C57BL/6 mice were utilized in muscle loss and muscle atrophy studies [60,61]. Factors such as mTOR, MyoD, Myostatin, MuRF1, FoxO3α, phospho-mTOR, phospho-FoxO3α, Bcl-2, caspase-3, PARP, IGF-1, Atrogin1, Akt, phospho-Akt, and four categories associated with oxidative stress, apoptosis, and muscle protein synthesis and degradation, prevalent in muscle atrophy, were analyzed. Furthermore, PCR and western blot analysis techniques were conducted to obtain more objective and differentiated data, allowing for cross-validation of gene and protein expression related to muscle loss and muscle atrophy factors [62]. Consequently, through a series of studies, our research team confirmed the muscle atrophy-related biological activity of AJHW, which utilized a pilot-scale hot water extraction method. This method ensures process stability and economic efficiency while using natural materials with a low risk of side effects, ultimately supporting the development of health functional foods and new drugs with muscle-loss-improving activities.

## 2. Results

### 2.1. Phytochemical Analysis

#### 2.1.1. Qualitative Analysis of AJHW

Using Thin Layer Chromatography (TLC), three representative active compounds in AJHW were identified by comparing the Rf values of *Alnus japonica* hot water extract (AJHW), 50% EtOH extract, and the standards oregonin, hirsutanonol, and hirsutenone (Figure 1).

#### 2.1.2. Quantitative Chromatographic Analysis of AJHW (HPLC)

First, the relationship between the content and peak area was calculated using the least-squares method (R^2^ value). After dividing the standard concentration into six levels, a calibration curve was obtained. For hirsutanonol, the equation Y = 13,167X − 385.69 (R^2^ = 0.9999) was derived, and for hirsutenone, it was Y = 15,625X + 2161.2 (R^2^ = 0.9999). Both substances exhibited excellent calibration curves (Figure 2A). The retention times of the standard compounds are illustrated in Figure 2B. Subsequent analysis revealed that the average content of hirsutanonol in the 50% EtOH extract was 3.96 ± 0.02 (*n* = 3), and for hirsutenone, it was 41.24 ± 1.09 (*n* = 3) (Figure 2C). Additionally, the average content of hirsutanonol in AJHW was determined to be 11.62 ± 0.15 (*n* = 3), and for hirsutenone, it was 73.02 ± 0.26 (*n* = 3) (Figure 2D).

#### 2.1.3. Qualitative Analysis of AJHW on LC-MS/MS

Mass spectrometry using LC-MS/MS was performed in the negative mode, confirming the mass values of each component in the AJHW sample as hirsutanonol: 344.8 *m*/*z* and hirsutenone: 326.9 *m*/*z* (Figure 3A). These were consistent with the mass values of each standard (Figure 3B,C).

### 2.2. Effects on Body Weight of Experimental Animals

The experimental animals were weighed weekly; all of the test groups exhibited normal weight changes, with continuous weight gain throughout the study period. Starting from the third week, body weights in the muscle atrophy test groups (G2, G3, G4, G5, G6) were lower than those in the normal group (G1) (Table 1).

### 2.3. Effects of Exercise Time and Exercise Capacity

To assess the impact of the test substance on the exercise performance of experimental animals, an incline of 10° and a speed of 10 m/min were sustained for the first 5 min. Subsequently, the speed increased by 1 m/min every minute to elevate exercise intensity, and the duration of exercise until exhaustion was recorded at a maximum speed of 25 m/min (Figure 4A). The exercise time until exhaustion in the normal control group (G1) was 42.4 ± 3.4 min, the longest among the test groups, while in the muscle atrophy control group (G2), it was significantly lower at 20.8 ± 0.8 min. In the muscle atrophy test groups (G2, G3, G4, G5, G6), the exercise time until exhaustion significantly increased in the 50 mg/kg AJHW treatment group (G4) at 24.8 ± 1.2 min, the 100 mg/kg AJHW treatment group (G5) at 27.6 ± 1.3 min, and the 200 mg/kg AJHW treatment group (G6) at 24.1 ± 1.1 min (Figure 4A).

Upon calculating the total exercise amount in the experimental animals, it was found that the exercise amount in the muscle atrophy control group (G2) significantly decreased to 945.4 ± 48.3 J from 2561.1 ± 233.0 J in the normal control group (G1). Following muscle atrophy induction, a significant increase in exercise amount was observed in the 50 mg/kg AJHW treatment group (G4) to 1205.7 ± 80.7 J and in the 100 mg/kg AJHW treatment group (G5) to 1390.8 ± 88.0 J (Figure 4B).

### 2.4. Effects on Grip Strength

To evaluate the effect of the test substance on the grip strength of the experimental animals, grip strength was measured using a BIO-GS3 Grip strength test for small animals (France). As indicated in Figure 4C, compared to 153.5 ± 8.2 g in the normal control group (G1), grip strength in the muscle atrophy control group (G2) significantly decreased to 90.3 ± 6.3 g. This significantly increased in the 50 mg/kg AJHW treatment group (G4) to 114.0 ± 5.3 g, in the 100 mg/kg AJHW treatment group (G5) to 118.5 ± 4.5 g, and in the 200 mg/kg AJHW treatment group (G6) to 115.9 ± 6.3 g, respectively (Figure 4C).

### 2.5. Effects on Fat Percentage and Lean Body Percentage

The influence of test substances on lean body mass and body fat percentage was analyzed by measuring the body composition components, and the findings are presented in Figure 5. Body fat percentage rose significantly from 17.2 ± 0.7% in the normal control group (G1) to 21.2 ± 0.8% in the muscle atrophy control group (G2). However, it decreased significantly to 18.1 ± 0.7% in the 100 mg/kg AJHW treatment group (G5) (Figure 5A). Similarly, lean body percentage decreased significantly from 82.8 ± 0.7% in the normal control group (G1) to 78.8 ± 0.8% in the muscle atrophy control group (G2), but it recovered to 81.9 ± 0.7% in the 100 mg/kg AJHW treatment group (G5) (Figure 5B).

### 2.6. Effects on Muscle Weight

Post-experimentation, the muscle weights of various body parts of the experimental animals were quantified, as illustrated in Table 2. Relative to the normal control group (G1), significant reductions in the weights of the quadriceps femoris (QF), gastrocnemius (GA), soleus (SOL), extensor digitorum longus (EDL), and tibialis anterior (TA) were observed in the muscle atrophy control group (G2). Conversely, the weights of the quadriceps femoris (QF), gastrocnemius (GA), soleus (SOL), and extensor digitorum longus (EDL) significantly increased in the 100 mg/kg AJHW treatment group (G5), while the gastrocnemius (GA) significantly increased in the 200 mg/kg AJHW treatment group (G6) (Table 2).

As the body weight (BW) of the experimental animals decreased due to muscle atrophy, the weight of each muscle group per 100 g of body weight (BW) was recalculated and is presented in Table 2. Compared with the normal control group (G1), the relative weights of the quadriceps femoris (QF), gastrocnemius (GA), soleus (SOL), and extensor digitorum longus (EDL) significantly decreased in the muscle atrophy control group (G2). Compared to the muscle atrophy control group (G2), the relative weights of the quadriceps femoris (QF) and soleus (SOL) significantly increased in the 100 mg/kg AJHW treatment group (G5) (Table 2).

### 2.7. Effects on Muscle Fiber Cross-Sectional Area of the Tibialis Anterior (TA)

Muscle size is controlled by intracellular signaling processes that drive anabolic or catabolic reactions within the muscles. When muscle loss and muscle atrophy manifests in this complex process, it results in a decrease in muscle protein, muscle size, and the number and diameter of muscle fibers. In this study, H&E staining was utilized using the tibialis anterior (TA) muscle to examine and gauge the cross-sectional area of muscle fibers to assess the extent of muscle loss. As indicated in Figure 6A, a significant reduction in the cross-sectional area of muscle fibers was noted in the muscle atrophy control group (G2) compared to the normal control group (G1), and an enhancement in the cross-sectional area of muscle fibers was observed due to the treatment of the test substance in the muscle atrophy treatment groups (G3, G4, G5, G6) relative to the muscle atrophy control group (G2). Following the measurement of the muscle fiber cross-sectional area (CSA), as depicted in Figure 6B, the muscle fiber cross-sectional area in the muscle atrophy control group (G2) significantly decreased to 286.7 ± 10.8 from 467.9 ± 10.6 in the normal control group (G1), and the muscle fiber cross-sectional area significantly increased in all test substance treatment groups (G3, G4, G5, and G6) (Figure 6).

### 2.8. Effects on Protein Expression in Gastrocnemius (GA)

#### 2.8.1. Effects on Muscle Degradation and Formation-Related Gene Expression (Protein)

Muscle protein synthesis is enhanced by regulating mTOR through the activation of PI3K and Akt signals via the Insulin-like Growth Factor-1 (IGF-1)/PI3K/Akt signaling pathway, a crucial anabolic mechanism involved in muscle formation and regeneration. Conversely, muscle protein degradation is controlled by a pathway that includes Forkhead box O (FoxO) and the ubiquitin-proteasome system. In skeletal muscle, FoxO consists of three isoforms (FoxO1, FoxO3α, and FoxO4), primarily located and activated in the nucleus. Upon phosphorylation by Akt, FoxO becomes inactivated and shifts from the nucleus to the cytoplasm. Conversely, reduced Akt activation triggers FoxO activation in the nucleus, enhancing the expression of muscle atrophy genes (Atrogin1 and MuRF1) that contribute to protein degradation. In this study, western blotting was conducted on muscle (gastrocnemius, GA) tissue lysates to assess the impact of test substance treatment on the activation of Akt, mTOR, and FoxO3α.

In the muscle atrophy control group (G2), p-Akt expression was significantly reduced compared to the normal control group (G1), whereas in the muscle atrophy test groups (G3, G4, G5, G6), all test substance treatment groups showed a significant increase in p-Akt expression compared to the muscle atrophy control group (G2). Akt expression showed no significant difference between the normal control group (G1) and the muscle atrophy control group (G2), although it was significantly reduced in the 100 mg/kg AJHW treatment group (G5) compared to the muscle atrophy control group (G2) (Figure 7B). The ratio of p-Akt/Akt, which measures Akt activity, revealed a significant decrease in the muscle atrophy control group (G2) compared to the normal control group (G1), with a significant increase in Akt activity observed in the test substance treatment groups (G3, G4, G5, and G6) (Figure 7B).

As depicted in Figure 7D, p-mTOR expression was significantly reduced in the muscle atrophy control group (G2) relative to the normal control group (G1). In contrast, p-mTOR expression significantly increased in the muscle atrophy test groups (G2, G3, G4, G5, G6) across all test substance treatment groups (G3, G4, G5, G6). However, mTOR expression itself significantly declined in the muscle atrophy control group (G2) compared to the normal control group (G1), showing no substantial differences in all test substance muscle atrophy groups (G3, G4, G5, and G6). Evaluating the activity of mTOR through the p-mTOR/mTOR ratio revealed a significant decrease in the muscle atrophy control group (G2) compared to the normal control group (G1) and a notable increase in the test substance treatment groups (G3, G4, G5, G6) (Figure 7D).

As illustrated in Figure 7F, p-FoxO3α expression was significantly reduced in the NAC group (G2) compared to the normal control group (G1). In the muscle atrophy test groups (G2, G3, G4, G5, G6), p-FoxO3α expression significantly increased in the 20 mg/kg AJHW treatment group (G3), 50 mg/kg AJHW treatment group (G4), and 200 mg/kg AJHW treatment group (G6) compared to the muscle atrophy control group (G2). FoxO3α expression increased significantly in the muscle atrophy control group (G2) compared to the normal control group (G1). Within the muscle atrophy test groups (G2, G3, G4, G5, G6), FoxO3α expression significantly rose in the 20 mg/kg AJHW treatment group (G3) compared to the muscle atrophy control group (G2), whereas it significantly decreased in the 50 mg/kg AJHW treatment group (G4), 100 mg/kg AJHW treatment group (G5), and 200 mg/kg AJHW treatment group (G6). Additionally, the p-FoxO3α/FoxO3α ratio significantly decreased in the muscle atrophy control group (G2) compared to the normal control group (G1) and increased significantly in all test substance treatment groups (G3, G4, G5, G6) (Figure 7F).

#### 2.8.2. Effects on Expression of Apoptosis Regulatory Proteins

Apoptosis is an important biological process associated with tissue regeneration, homeostasis, maintenance, and growth. Muscle cell apoptosis induced by chronic inflammation, including oxidative stress, is a primary cause of muscle loss and muscle atrophy. During apoptosis, the activity of Bax increases (a pro-apoptotic factor) while the expression of Bcl-2 (an anti-apoptotic factor) is relatively suppressed. A disrupted balance between these factors leads to mitochondrial dysfunction and the release of cytochrome c from the inner mitochondrial membrane into the cytoplasm, subsequently down-regulating various genes and inducing apoptosis. In this study, western blotting was conducted using gastrocnemius (GA) tissue lysates to examine the impact of the test substance on Bcl-2 and Bax expression. As shown in Figure 8B, the Bcl-2 expression was significantly lower in the muscle atrophy control group (G2) compared to the normal control group (G1). Conversely, the decreased Bcl-2 expression in the muscle atrophy control group (G2) significantly increased in the 50 mg/kg AJHW treatment group (G4), 100 mg/kg AJHW treatment group (G5), and 200 mg/kg AJHW treatment group (G6). There was no significant difference in Bax expression between the normal control group (G1) and muscle atrophy control group (G2). However, in the muscle atrophy test groups (G3, G4, G5, G6), Bax expression significantly decreased in the 100 mg/kg AJHW treatment group (G5) and 200 mg/kg AJHW treatment group (G6) compared to the muscle atrophy control group (G2) (Figure 8C).

### 2.9. Effects on mRNA Expression in Muscle

#### 2.9.1. Effects on Muscle Degradation and Formation-Related Gene Expression (mRNA)

An important mechanism implicated in muscle atrophy-related metabolism involves the ubiquitin-proteasome signaling pathway. When a muscle atrophy signal occurs, ubiquitin, ubiquitin-conjugating enzyme (E2), ubiquitin-protein ligases (E3), and proteasome factors bind to skeletal muscle proteins, facilitating muscle atrophy progression. During this process, two specific E3 ubiquitin ligases, namely muscle atrophy F-box/Atrogin1 and Muscle Ring Finger-1 (MuRF1), which are expressed only in muscles, are crucial in activating protein ligases and accelerating muscle atrophy. Additionally, myostatin, a member of the TGF-β family, acts as an inhibitor, uniquely expressed in muscles, and obstructs skeletal muscle growth by reducing Akt/mTOR/p70S6K signaling. Conversely, exercise promotes muscle regeneration and generation, activating myogenin regulatory factors during this phase. These factors are muscle-specific basic helix–loop–helix transcription factors, which include MyoD, Myf5, myogenin, and Mrf4, expressed in satellite cells. MyoD identifies the myoblasts, precursors of muscle formation, and myogenin encourages muscle differentiation. This study examined the impact of muscle atrophy induction and test substance treatment on the expression of muscle formation-related genes (MyoD, myogenin) and muscle atrophy-related genes (myostatin, Atrogin1, MuRF1) in muscle (SOL), with the findings presented in Figure 9. The mRNA expression of muscle formation-related genes MyoD, myogenin, and IGF-1 was significantly lower in the muscle atrophy control group (G2) than in the normal control group (G1). However, the mRNA expression of MyoD and IGF-I, reduced in the muscle atrophy control group (G2), was significantly elevated in both the 50 mg/kg and 100 mg/kg AJHW treatment groups (G4, G5), while myogenin mRNA expression, also reduced in the muscle atrophy control group (G2), substantially increased in all of the treatment groups (G4, G5, G6). The mRNA expression of muscle atrophy-related genes, myostatin, Atrogin1, and MuRF1, showed a significant increase in the muscle atrophy control group (G2) compared to the normal control group (G1). In the muscle atrophy test groups (G2, G3, G4, G5, G6), mRNA expression of myostatin, Atrogin1, and MuRF1 was significantly reduced in the 100 mg/kg and 200 mg/kg AJHW treatment groups (G5, G6) compared to the muscle atrophy control group (G2). The results are displayed in Figure 9.

#### 2.9.2. Effects on Inflammation-Related Gene Expression

The mRNA levels of IL-1β, IL-6, and TNF-α, inflammation-related genes in the soleus muscle, were investigated. As indicated in Figure 10, the mRNA expression of IL-1β, IL-6, and TNF-α in the soleus muscle was significantly increased in the muscle atrophy control group (G2) compared to that in the normal control group (G1). In the muscle atrophy test groups (G2, G3, G4, G5, G6), the expression of TNF-α mRNA was significantly decreased in the 100 mg/kg AJHW treatment group (G5) compared to the muscle atrophy control group (G2), and the expression of IL-1β and IL-6 mRNA showed a decreasing trend with the treatment, but the differences were not significant (Figure 10).

#### 2.9.3. Effects on Antioxidant-Related Gene Expression

We investigated the mRNA levels of the antioxidant genes SOD2, catalase, and GPx1 in the soleus muscle. As indicated in Figure 11, the expression of SOD2 catalase and GPx1 mRNA was significantly reduced in the muscle atrophy control group (G2) compared to the normal control group (G1). In the muscle atrophy test groups (G2, G3, G4, G5, G6), relative to the muscle atrophy control group (G2), catalase mRNA expression significantly increased in the 50 mg/kg AJHW treatment group (G4), 100 mg/kg AJHW treatment group (G5), and 200 mg/kg AJHW treatment group (G6). Likewise, GPx1 mRNA expression significantly rose in the 100 mg/kg AJHW treatment group (G5). SOD2 mRNA expression showed a tendency to increase after test substance treatment compared to the muscle atrophy control group (G2), although the difference was not statistically significant (Figure 11).

### 2.10. Effects on Serum Glutathione Content and Antioxidant Enzyme Activity

In a normal biological state, the human body maintains a balance between the production of free radicals and the activity of the antioxidant defense system, preventing tissue or cell damage. However, due to physical stress (muscle atrophy induction), excessive production of active oxygen diminishes the function of the antioxidant system, disrupting this balance and leading to excessive oxidative stress. The body contains key antioxidant substances and enzyme systems, such as glutathione, SOD, catalase, and GPx, that comprise the antioxidant system. Oxygen free radicals are converted into hydrogen peroxide (H_2_O_2_) by SOD, which catalase and GPx then convert into water, detoxifying it. To assess the impact of muscle atrophy induction and test substance treatment on oxidative stress and the antioxidant system, we measured the content of glutathione and the activities of enzymes such as SOD, catalase, and GPx in serum, with the results presented in Table 3. Compared to the normal control group (G1), the glutathione content in serum was significantly reduced in the muscle atrophy control group (G2), but significantly increased in all test substance treatment groups (G3, G4, G5, G6) compared to the muscle atrophy control group (G2) (Table 3). Serum SOD, catalase, and GPx activities were significantly lower in the muscle atrophy control group (G2) than in the normal control group (G1). Serum SOD activity was significantly higher in the 200 mg/kg AJHW treatment group (G6) compared to the muscle atrophy control group (G2), and serum catalase activity was considerably enhanced in the 100 mg/kg AJHW treatment group (G5) compared to the muscle atrophy control group (G2). Serum GPx activity was significantly higher in the 50 mg/kg, 100 mg/kg, and 200 mg/kg AJHW treatment groups (G4, G5, G6) than in the muscle atrophy control group (G2) (Table 3).

## 3. Discussion

In our previous studies, we validated the biological activity of *Ulmus macrocarpa* extract against muscle atrophy and conducted preclinical in vitro and in vivo studies [48,49]. Based on our in vitro findings of AJHW against muscle loss and muscle atrophy, our goal was to confirm the efficacy of AJHW in ameliorating sarcopenia through rigorous in vivo studies [61]. Particularly, in vitro experiments are indispensable in preclinical studies for validating specific bioactive responses in controlled conditions, while in vivo studies are crucial for evaluating broader external factors and individual variations.

In the grip strength test, the muscle atrophy control group (G2) exhibited a 58% reduction in grip strength compared to the normal control group (G1), while the AJHW 100 mg/kg treatment group (G5) showed about 20% more grip strength recovery than the muscle atrophy control group (G2). Compared to a prior *Ulmus macrocarpa* extract trial, which resulted in 14% grip strength recovery with 200 mg/kg treatment, AJHW demonstrated enhanced efficacy [49]. Additionally, the 100 mg/kg treatment group (G4) displayed more than 45% improved exercise capacity compared to the muscle atrophy control group (G2). Conversely, an earlier *Ulmus macrocarpa* experiment identified approximately 43% improvement in the 100 mg/kg treatment group compared to the control group [49], and a *Alnus japonica* 50% EtOH extract treatment at 200 mg/kg was reported to increase exercise capacity by about 35% in comparison with the control group [47]. Therefore, the results indicate that AJHW is considerably more effective than other extracts in enhancing muscle strength and exercise capacity.

The results of muscle weighing indicated that all regional muscles, except the tibialis anterior (TA), exhibited statistically significant increases in weight in the AJHW treatment group compared to the muscle atrophy control (G2). Specifically, the quadriceps’ (QF) weight increased by approximately 19%, the gastrocnemius’ (GA) by 11%, the soleus’ (SOL) by 10%, and the extensor digitorum longus’ (EDL) by 15% in the 100 mg/kg AJHW treatment group (G5) compared to the muscle atrophy control group (G2). Furthermore, histological measurements of muscle fibers revealed that the 50 mg/kg AJHW treatment group (G4) saw a 47% increase in muscle fiber size compared with the muscle atrophy control group (G2), while the 100 mg/kg *Ulmus macrocarpa* extract treatment group experienced a 50% increase in muscle fiber size compared to the control group [49]. The 125 mg/kg *Schisandra chinensis* extract treatment group saw a 25% increase in muscle fiber size compared to the control group [63], and the 40 mg/kg *Curcuma longa* extract treatment group exhibited a 42% increase compared to the control group [64]. This suggests that AJHW not only mitigates muscle weight loss due to muscle loss but also effectively enhances muscle fiber size at low concentrations compared with other extracts.

Protein expression analysis of factors associated with muscle atrophy revealed that p-Akt protein expression increased by over 70% in the 50 mg/kg AJHW treatment group (G4) compared with the control group (G2), while the *Ulmus macrocarpa* extract exhibited an approximate 57% increase at 100 mg/kg treatment compared to the control group [49]. Expression of the p-mTOR factor increased by more than 300% following 100 mg/kg AJHW treatment (G5) relative to the muscle atrophy control group (G2), while a 54% increase was noted in the *Ulmus macrocarpa* extract experiment compared to the control [49]. Additionally, p-Foxo3α expression surged by over 70% in the 20 mg/kg AJHW treatment group (G3) compared to the muscle atrophy control group (G2), underscoring AJHW’s heightened activity at lower doses compared to the *Ulmus macrocarpa* extract. Analysis of apoptosis factors Bax and Bcl-2 demonstrated that Bax decreased by approximately 20% in the 100 mg/kg AJHW treatment group (G5) relative to the muscle atrophy control group (G2), while Bcl-2 expression increased by more than 30% at the same concentration. Thus, AJHW significantly regulates protein expression related to muscle synthesis and degradation at low concentrations and also moderates apoptosis.

Subsequent RT-PCR analysis examined the mRNA expression of Atrogin1 and MuRF1 factors involved in muscle degradation and found that Atrogin1 mRNA expression was reduced by more than 50% in the AJHW 100 mg/kg treatment group (G5) compared to the muscle atrophy control group (G2). In contrast, an approximately 28% decrease was observed in the 200 mg/kg *Ulmus macrocarpa* extract treatment [49], and a 50% decrease in mRNA expression was observed in the 500 mg/kg *Schisandra chinensis* extract treatment [63]. Moreover, treatment with 200 mg/kg *Alnus japonica* 50% EtOH extract resulted in an approximately 40% lower mRNA expression compared to the muscle atrophy control [47]. For MuRF1, the 100 mg/kg treatment (G5) with AJHW showed a decrease in mRNA expression of over 30% compared to the muscle atrophy control, similar to the approximately 30% decrease observed in the 125 mg/kg treatment with *Schisandra chinensis* extract [63]. Therefore, AJHW was concluded to have comparable activity at lower concentrations and potent effects on the mRNA expressions of Atrogin1 and MuRF1, muscle degradation factors. The mRNA expression of myostatin, a muscle growth inhibitory factor, was decreased by more than 50% in the 200 mg/kg AJHW treatment group (G6) compared to the muscle atrophy control group (G2), showing a higher inhibitory effect than the approximately 45% decrease in the 250 mg/kg *Schisandra chinensis* extract treatment group [63]. Consequently, AJHW regulated the mRNA gene expression of muscle protein synthesis and degradation factors and demonstrated high activity at lower concentrations compared to other extracts.

For the analysis of antioxidant-related factors, the mRNA expression of SOD, catalase, and GPx, as well as antioxidant enzyme activities in serum, were measured. The results revealed no significant change in SOD mRNA expression following AJHW treatment. However, SOD activity in serum dropped by approximately 35% in the muscle atrophy control group (G2) compared to the normal control group (G1). In contrast, in the 200 mg/kg AJHW treatment group (G6), SOD activity was restored to levels similar to those in the normal control group (G1). Catalase mRNA expression decreased by about 60% in the muscle atrophy control group (G2) compared to the normal control group (G1), but it increased by approximately 62% in the AJHW 50 mg/kg treatment group (G4) compared to the muscle atrophy control group. Similarly, catalase activity in serum decreased by about 35% in the muscle atrophy control group (G2) and increased by approximately 43% in the AJHW 100 mg/kg treatment group (G5). For GPx, a decrease in mRNA expression by roughly 43% was noted in the muscle atrophy control group (G2) compared to the normal control group (G1), with an increase of about 49% in the AJHW 100 mg/kg treatment group (G5) compared to the muscle atrophy group (G2). Analysis of GPx activity in serum indicated a decrease of approximately 30% in the muscle atrophy control group, but it was restored to normal levels in the AJHW 100 mg/kg treatment group (G5). Even so, the enzymes obtained from the collected serum were not extracted from specific muscles. Therefore, this presents a limitation in fully reflecting the specificity of enzymes expressed in various sources, such as erythrocytes and muscle tissues.

In this experiment, the efficacy of AJHW was assessed by comparing it with results from other studies that used different natural products; prior to this analysis, AJHW and the *Alnus japonica* 50% EtOH extract were comparatively analyzed through phytochemical analysis to determine differences between the extracts. Phytochemical analysis of oregonin in AJHW and *Alnus japonica* 50% EtOH extract was previously conducted. However, phytochemical analysis of hirsutanonol and hirsutenone, reported to exhibit excellent anticancer properties along with significant biological activities such as anti-inflammation, anti-toxicity, and hepatoprotection, was inadequate, warranting further analysis [56,65,66].

The experimental results indicated that AJHW contained higher levels of hirsutanonol and hirsutenone, aglycone forms of oregonin, compared to the *Alnus japonica* 50% EtOH extract. Subsequent muscle atrophy biological activity experiments revealed that AJHW significantly enhanced grip strength recovery, exercise capacity, muscle weight, muscle fiber size, and protein expression (p-Akt, p-mTOR, p-Foxo3α) relative to the muscle atrophy control group (G2). Additionally, it reduced the mRNA expression of muscle degradation-related factors such as Atrogin1 and MuRF1. Moreover, AJHW was found to promote muscle growth by suppressing Myostatin expression

Additionally, when compared to previously reported experiments, AJHW exhibited higher biological activity at lower concentrations than the *Ulmus macrocarpa* extract and *Schisandra chinensis* extract. Furthermore, the significant sarcopenia improvement effect of the *Curcuma longa* extract, marked by the diarylheptanoid series compound curcumin, has been previously reported [64], suggesting that diarylheptanoid components may be more effective in ameliorating sarcopenia than other series compounds [64].

Meanwhile, comparative analysis of AJHW and *Alnus japonica* 50% EtOH extract confirmed the superior biological activity of AJHW [47]. This higher activity is attributed to the synergistic effects of the elevated levels of hirsutanonol and hirsutenone in AJHW, combined with oregonin. Consequently, considering the levels of hirsutanonol, hirsutenone, and oregonin, the hot water extraction method could yield more active substances than the 50% EtOH extraction in future pilot-scale processes. Additionally, the hot water extraction method is considered more suitable for production, given its advantages in safety, material stability, cost efficiency, and environmental friendliness.

In conclusion, the present study reports the results of a comparative analysis of the biological activity of AJHW in various mechanisms of muscle loss and muscle atrophy, including protein synthesis and degradation, apoptosis, and antioxidant activity. Compared to the previously documented *Alnus japonica* 50% EtOH extract, *Ulmus macrocarpa* extract, and *Schisandra chinensis* extract, AJHW demonstrated consistently high biological activity corresponding to its concentration [47,49,63]. Specifically, AJHW exhibited a superior effect compared to the *Alnus japonica* 50% EtOH extract [47] due to the synergistic actions of hirsutanonol and hirsutenone, the aglycone forms of oregonin identified through phytochemical analysis. Thus, AJHW holds promise as a potent ingredient for developing functional products or pharmaceuticals aimed at preventing and ameliorating muscle loss.

However, this study has certain limitations that must be considered. First, only one muscle atrophy model induced by dexamethasone was tested; therefore, future studies should validate the effects of AJHW in other models, such as cancer cachexia and disuse-induced atrophy. Second, although phytochemical analysis identified the presence of oregonin, hirsutanonol, and hirsutenone, the specific contributions of each compound remain unclear, necessitating further studies using purified substances. Third, the exact molecular mechanisms underlying the effects of AJHW were not fully elucidated and should be explored through mechanistic studies using knockout models. Finally, potential long-term toxicity was not evaluated in this study and should be addressed in future investigations to ensure safety for therapeutic applications.

## 4. Materials and Methods

### 4.1. Plant Extract Materials and Method

#### 4.1.1. Plant Extract Materials

This study utilized pilot scale *Alnus japonica* hot water extract (AJHW). The *Alnus japonica* stem, including barks, for this research was sourced from Seoul Yakryeong Market. Additionally, another material was sourced from research cultivation filed at Gangwon State Forest Science Institute (24, Hwamokwon-gil, Chuncheon-si, Gangwon State, 24207, Republic of Korea) and certified by Professor Choi (Department of Forest Biomaterials Engineering, Kangwon National University). The materials were cleaned and washed to eliminate impurities, after which two materials were mixed and used as experimental materials.

#### 4.1.2. Plant Pilot-Scale Extraction Method

*Alnus japonica* hot water extract powder was produced on a pilot scale using the following process: 300 kg of raw material was extracted with 3000 L of distilled water at 100 ± 5 °C for 4 h before concentration. The concentrated procedure employed a natural circulation technique, and 136.55 kg of *Alnus japonica* concentrate was subsequently obtained by concentrating further at 60 ± 5 °C (yield 45.52%). Additionally, 10% dextrin was added to 300 mL of raw material, then freeze-dried to yield 8.56 g of *Alnus japonica* hot water extract powder (yield 81.52%, Lot No. 230901, Lot No. DJTF-11872). This extraction process was conducted by ChuncheonBio Co., LTD (Chuncheon-si, Republic of Korea) and DanjoungBio Co., LTD (Wonju, Republic of Korea). Additionally, this study utilized *Alnus japonica* 50% EtOH extract, prepared similarly to previous research for producing ethanol extracts of the alder tree. Specifically, 500 g of dried bark was extracted with 5 L of 50% ethanol for 7 h at 80 ± 5 °C, sterilized, and evaporated under reduced pressure at 65 °C. The resulting concentrate was mixed at a 1:1 (*w*/*w*) ratio with maltodextrin DE20 and was spray-dried using a pilot-scale spray dryer, yielding 20%.

### 4.2. Phytochemical Analysis Methods

#### 4.2.1. Standard Material for Phytochemical Analysis

In this study, the standard compounds oregonin, hirsutanonol, and hirsutenone were utilized. Oregonin was obtained using the same methods as previous studies and through in vitro mechanism research [41,51,56,57,61]. Hirsutanonol and hirsutenone were acquired during previous research and have been stored in our laboratory; the samples used in this study were sourced from these purified and stored materials [56,57,61,67] (Figure 12).

Oregonin (A) Brown amorphous powder, Negative LC-MS/MS: *m*/*z* 477 [M − H] −

1H-NMR (700 MHz, MeOH-d4): 6.657 (H-2″, 1H, d, *J* = 2.8 Hz), 6.645 (H-5′, 1H, d, *J* = 2.8 Hz), 6.616 (H-5″, 1H, d, *J* = 2.1 Hz), 6.607 (H-2′, 1H, d, *J* = 2.1 Hz), 6.492 (H-6″, 1H, d, *J* = 2.1 Hz), 6.470 (H-6′, 1H, d, *J* = 2.1 Hz), 4.217 (xyl-1, 1H, d, 7.7 = Hz), 3.865 (H-5, 1H, d, *J* = 5.6 Hz), 3.312 (xyl-1, 1H, d, *J* = 1.4 Hz), 3.310 (xyl-5e, 1H, d, *J* = 2.1 Hz), 3.303 (xyl-4, 1H, d, *J* = 4.2 Hz), 3.168, 3.124 (2H, m, xyl-3, 5a), 2.85–2.45 (8H in total, H-1,2,4,7), 1.778–1.702 (2H in total, m H-6).

13C-NMR (150 MHz, MeOH-d4): 146.27 (C-3″), 146.18 (C-3′), 144.56 (C-4′), 144.29 (C-4″), 135.28 (C-1′), 133.17 (C-1″), 120.82 (C-6′), 120.72 (C-6″), 116.74 (C-5″),116.66 (C-5′), 115.51 (C-2″), 115.40 (C-2′), 104.44 (Xyl-1), 78.03 (Xyl-3), 76.47 (C-5), 75.24 (Xyl-2), 71.40 (Xyl-4), 67.09 (Xyl-5), 28.66 (C-4), (C-2), (C-6), (C-7), 30.23 (C-1,C-3) [41,51,56,57].

Hirsutanonol (B) Brown Oil, Negative LC/MS-MS: *m*/*z* 345 [M − H]−

-1H-NMR (700 MHz, MeOH-d4): δ = 6.61–6.69 (4H in total, m, H-2′,2″,5′5″), 6.49–6.53 (2H in total, m, H-6′,6″), 4.02 (1H, m, H-5), 2.46–2.75 (8H in total, m, H-1,2,4,7), 1.66–1.70 (2H in total, m, H-6).

13C-NMR (150 MHz, MeOH-d4): δ = 210.65 (C-3), 144.78 (C-3″), 144.72 (C-3′), 143.06 (Cc-4″), 142.84 (C-4′), 133.58 (C-1″), 132.69 (C-1′), 119.26 (C-6″), 119.13 (C-6′),115.16 (C-5″), 115.08 (C-5′), 114.97 (C-2″), 114.92 (C-2′), 66.93 (C-5), 49.86 (C-4), 44.96 (C-2), 39.02 (C-6), 30.76 (C-7), 28.66 (C-1) [56,57,67].

**Hirsutenone (C)** Brown Oil, Negative LC/MS-MS: *m*/*z* 326 [M − H] −

1H-NMR (700 MHz, MeOH-d4): δ = 6.74–6.78 (4H in total, m, H-2′,2″,5′5″), 6.50–6.57 (2H in total, m, H-6′,6″), 5.97 (1H, D, *J* = 16.0 HzH-4), 2.34–2.71 (8H in total, m, H-1,2,6,7)

13C-NMR (150 MHz, MeOH-d4): δ = 201.48 (C-3), 147.87 (C-5), 144.81 (C-3″), 144.76 (C-3′), 143.13 (C-4″), 143.10 (C-4′), 132.64 (C-1″), 132.46 (C-1′), 130.19 (C-4), 119.30 (C-6″), 119.19 (C-6′), 115.12 (C-5″,5′), 114.95 (C-2″,2), 41.33 (C-2), 34.22 (C-6), 33.40 (C-7), 29.54 (C-1) [56,57,67].

#### 4.2.2. Qualitative Analysis of AJHWE (TLC)

Thin layer chromatography was used to identify components in *Alnus japonica* hot -water extract (AJHW) and *Alnus japonica* 50% EtOH extract. A standard and a sample were each weighed at 3.5 mg, dissolved in 1 mL of methanol, and prepared as 3500 ppm solutions. The samples were spotted onto silica gel plates and developed using a mobile phase of chloroform/methanol/water (CMW) in a 70:30:4 ratio. After the plate was completely dry, it was observed under a 254 nm UV lamp, and analysis was conducted using three different coloring reagents: 10% H_2_SO_4_, ρ-anisaldehyde H_2_SO_4_, and FeCl_3_.

#### 4.2.3. Quantitative Chromatographic Analysis of AJHW (HPLC) Method

HPLC analysis of *Alnus japonica* hot water extract (AJHW) utilized a Waters 2695 HPLC system and Waters 2487 HPLC UV/Vis Detector (WATERS Corporation, Milford, MA, USA). The mobile phases were 1% acetic acid in distilled water (A) and acetonitrile (B), with the process conducted at a 1 mL/min flow rate for 40 min. The gradient program was set as follows: 0–0 min, 10% B; 0–12 min, 25% B; 12–24 min, 40% B; 24–25 min, 10% B; 25–40 min, 10% B. Quantitative analysis for the *Alnus japonica* hot water extract (AJHW) and 50% EtOH extract involved hirsutanonol and hirsutenone standards previously isolated from *Alnus japonica*. Both compounds were measured at 1 mg, dissolved in 1 mL of methanol, and diluted to concentrations of 100, 50, 25, 10, 5, and 1 ppm.

#### 4.2.4. Measurement of the Molecular Weight of AJHW (LC-MS/MS)

For the LC-MS/MS analysis of AJHW, the AB SCIEX (QTRAP 4500, Billerica, MA, USA) system was utilized. The column configuration included a HECTOR C18 column (5 μm) paired with a Phenomenex KJ0-4282 guard column. The analyses used mobile phases of 1% acetic acid in water (A) and acetonitrile (B), a wavelength of 280 nm, and a flow rate of 1 mL/min for 40 min. The gradient was set as follows: 0–0 min, 10% B; 0–12 min, 25% B; 12–24 min, 40% B; 24–25 min, 10% B; and 25–40 min, 10% B.

### 4.3. Ethical Statement and Animals

The experimental animal guidelines adhered to the regulations for the use and care of experimental animals as set by the Korea Food and Drug Administration. Approval for the use of experimental animals was granted by the Institutional Animal Care and Use Committee of Hallym University, which was approved by the National Agricultural Products Quality Management Service under the Ministry of Agriculture, Food and Rural Affairs of Korea (approval number: Hallym 2023-33).

Eight-week-old male C57BL/6 mice free of specific pathogens were acquired from Dooyeol Biotech Co. Ltd. in Seoul, Republic of Korea. Following a one-week quarantine and adaptation period, only healthy animals without any weight loss were selected for the experiment. The experimental animals were housed in a breeding environment maintained at a temperature of 23 ± 3 °C, a relative humidity of 50 ± 10%, a ventilation rate of 10–15 times/hour, a lighting cycle of 12 h (08:00–20:00), and an illumination intensity of 150–300 Lux. During the adaptation period, the animals had free access to solid feed from Cargill Agri Purina Co. Ltd., Seoul, Republic of Korea and water.

### 4.4. Experimental Design and Treatment

After a one-week adaptation period, healthy animals were selected and assorted into six test groups using the randomized block method. The groups consisted of: (G1) normal control group, (G2) muscle atrophy control group, (G3) muscle atrophy + 20 mg/kg AJHW treatment group, (G4) muscle atrophy + 50 mg/kg AJHW treatment group, (G5) muscle atrophy + 100 mg/kg AJHW treatment group, and (G6) muscle atrophy + 200 mg/kg AJHW treatment group, as depicted in Figure 13. Also, a total of ten animals was utilized in each group. The test substance was dissolved in drinking water and administered orally once daily for four weeks. To induce muscle loss (atrophy) after two weeks of treatment, dexamethasone (Merck & Co., Inc., Kenilworth, NJ, USA) was injected intraperitoneally at a dose of 5 mg/kg body weight (BW) once a day. Throughout the entire test period, the animals were fed solid feed from Cargill Agri Purina Co. Ltd., Republic of Korea, Seoul, and had free access to food and water. The body weight of the animals was consistently monitored once a week at a fixed time starting from the date of the test substance treatment (Figure 13).

### 4.5. Measurement of Exercise Capacity

One and two days before measuring exercise capacity, the mice were subjected to training sessions involving running speeds of 10 m/min over a 5-min period with a 10° slope. Three days prior to the test’s conclusion, a treadmill exercise was conducted to assess endurance performance in small animals (MZ-2021V152; HUAYON Biotech Co., Ltd., Shenzhen, China). Endurance was tested by initiating exercise at a 10° slope and a 10 m/min speed for 5 min. The speed increased by 1 m/min each minute to intensify the exercise, measuring the duration of exercise until exhaustion at a maximum speed of 25 m/min. Exhaustion was defined as the point where the animal could not maintain running for more than 10 s, leaning toward the rear of the treadmill. The exercise load for the animals was quantified using the formula:Exercise capacity (J, kg m^2^ s^−2^) = body weight (kg) × speed (m/s) × time (s) × grade × 9.8 m/s^2^(1)

### 4.6. Measurement of Grip Strength

A grip strength test was conducted one day prior to the test’s end. One day before measuring grip strength, the mice underwent 5 trials of grip strength measurements to acclimate to the grip strength tester. This test measured grip strength in grams using a small animal grip strength tester (BIO-G53; Bioseb, Chaville, France). After coaxing the animals to grip the T-bar with their forearms, the tail was uniformly pulled at 2 cm/s until the forearms released the bar. Five readings were taken per animal, and average values were computed.

### 4.7. Measurement of Lean Body Percentage and Fat Percentage

One day prior to the test’s conclusion, the animals were anesthetized and evaluated for body composition using dual-energy X-ray absorptiometry (DEXA, GE Lunar; GE Healthcare Technologies, Inc., Chicago, IL, USA), assessing lean body mass and body fat percentage.

### 4.8. Blood and Tissue Collection

Prior to sacrifice, the animals were anesthetized using a tribromoethanol-tertiary amyl alcohol mixture, and orbital blood was drawn. This blood was collected in a serum separator tube (Becton, Dickinson and Company, Franklin Lakes, NJ, USA), left at room temperature for 30 min, then centrifuged at 5000 rpm for 10 min to separate the serum, which was subsequently stored at −70 °C. Following blood collection, animals were euthanized, and the key muscles—quadriceps femoris (QF), gastrocnemius (GA), soleus (SOL), extensor digitorum longus (EDL), and tibialis anterior (TA)—were excised, cleansed with cold saline, dried with filter paper, and weighed. The total RNA was isolated from the soleus muscle (SOL) for real-time PCR, parts of the gastrocnemius muscle (GA) were prepared for western blotting after protein isolation, and a portion of the tibialis anterior muscle (TA) was fixed in 4% paraformaldehyde (PFA) and embedded in paraffin for tissue immunostaining. The remaining tissues were conserved at −70 °C.

### 4.9. Measurement of Antioxidant Enzyme Activity and Glutathione Content in Serum

The activities of antioxidant enzymes in the serum, including superoxide dismutase (SOD, Cayman Chemical, Ann Arbor, MI, USA), catalase (Cayman Chemical, Ann Arbor, MI, USA), and glutathione peroxidase (GPx, Cayman Chemical, Ann Arbor, MI, USA), were quantified using their respective kits as per the manufacturer’s instructions. The glutathione content in serum was assessed using a glutathione assay kit (Cayman Chemical, Ann Arbor, MI, USA) following the prescribed method.

### 4.10. Hematoxylin and Eosin Stain

The tibialis anterior (TA) muscle was fixed with 4% PFA, embedded in O.C.T compound (Miles scientific, Newark, DE, USA) for freezing, and sectioned into 6 μm slices at −25 °C using Thermo Scientific equipment. These sections were air-dried, washed five times with PBS for 3 min each, and stained with Accustain^®^ Hematoxylin and Eosin Stains (Sigma-Aldrich Co., St.Louis, MO, USA) following the manufacturer’s instructions to examine the histomorphology of the tibialis anterior (TA). Histological alterations were examined under an optical microscope (Carl Zeiss, Oberkochen, Baden-Wurttemberg, Germany). For CSA analysis, we applied the minimal Feret’s diameter method for calculation and quantified the results using ImageJ software (Version 1.54h).

### 4.11. Western Blot Analysis

To examine protein expression in the gastrocnemius muscle (GA), lysis buffer (20 mmol/L HEPES, pH 7.5, 150 mmol/L NaCl, 1% Triton X-100, 1 mmol/L EDTA, 1 mmol/L EGTA, 100 mmol/L NaF, 10 mmol/L sodium pyrophosphate, 1 mmol/L Na_3_VO_4_, 20 μg/mL aprotinin, 10 μg/mL antipain, 10 μg/mL leupeptin, 80 μg/mL benzamidine HCl, 0.2 mmol/L PMSF) was added and homogenized. Following centrifugation at 12,000 rpm for 10 min, the supernatant was collected to obtain muscle tissue lysates. The protein content in these lysates was measured using a BCA protein assay kit (Thermo Scientific, Waltham, MA, USA).

The proteins (50 μg) were separated by 10% sodium dodecyl sulfate polyacrylamide gel electrophoresis (SDS-PAGE) and then transferred to a polyvinylidene difluoride membrane (Millipore, Billerica, MA, USA). The membrane was subsequently blocked for 1 h in a 5% skim milk-TBST solution (20 mM/L Tris-HCl, pH 7.5, 150 mM/L NaCl, 0.1% Tween 20). Antibodies targeted for measurement were introduced, and the mixture was agitated at 4 °C for 16 h or at room temperature for 1 h. Details about the utilized antibodies are provided in Table 4. Following this, horseradish peroxidase (HRP)-linked anti-rabbit IgG and anti-mouse IgG were added and agitated for an additional hour. The protein bands were visualized using the enhanced chemiluminescence method with Luminata^TM^ Forte Western HRP Substrate (Millipore, Billerica, MA, USA). Protein expression levels were quantified with the ImageQuant^TM^ LAS 500 imaging system (GE Healthcare Bio-Science AB, Danderyd, Sweden).

### 4.12. Real-Time Polymerase Chain Reaction (Real-Time PCR)

The total RNA was isolated from SOL muscle tissue using TRIzol Reagent (Thermo Scientific, Waltham, MA, USA) and was quantified with a BioSpec-nano microvolume spectrophotometer (SHIMADZU, Kyoto, Japan). RNA displaying an OD260/280 value of 1.8 or higher was selected for the experiments. cDNA was synthesized from 2 μg of total RNA using the HyperScript^TM^ RT master mix kit (GeneAll Biotechnology Co., Ltd., Seoul, Republic of Korea), and real-time PCR was carried out using a Rotor-Gene 300 PCR machine (Corbett Research, Mortlake, Australia) and a Rotor-Gene^TM^ SYBR Green kit (Qiagen, Germantown, MD, USA). The primers used are listed in Table 5. Quantitative analysis of gene expression was conducted with the Rotor-Gene 6000 Series System Software version 6 (Corbett Research, Mortlake, New South Wales, Australia).

### 4.13. Statistical Analysis

All analysis values are expressed as the mean ± SEM. The results were analyzed using GraphPad Prism 5.0 (GraphPad Software). Student’s *t*-test and one-way analysis of variance (ANOVA) were utilized to compare differences between the test substance administration and control groups. Statistical significance was set at *p* < 0.05.

## 5. Conclusions

Sarcopenia, a muscle disease associated with aging, leads to a decline in muscle strength, volume, and function, impairing quality of life. This study evaluated the efficacy of *Alnus japonica* hot water extract (AJHW) in mitigating muscle loss and muscle atrophy. AJHW demonstrated superior biological activity compared to *Alnus japonica* 50% EtOH extract, with elevated levels of hirsutanonol and hirsutenone, compounds known for their synergistic effects with oregonin [56,65,66].

Experimental results showed AJHW significantly improved grip strength, muscle weight, fiber size, and protein expression (p-Akt, p-mTOR, p-Foxo3α), while suppressing muscle degradation markers (Atrogin1, MuRF1) and promoting muscle growth by inhibiting myostatin expression. AJHW also showed a trend of restoring antioxidant enzyme activity (SOD, Catalase, GPx), which may have contributed to the improvement of muscle atrophy. Consequently, our research team conducted a comprehensive analysis of all four categories known to contribute to the development of muscle loss.

Compared to other natural extracts, including *Ulmus macrocarpa* and *Schisandra chinensis*, AJHW showed higher efficacy even at lower concentrations [49,63]. Its extraction method has proven to be advantageous in terms of safety, cost efficiency, and environmental sustainability. These findings position AJHW as a promising resource for the development of functional products and pharmaceuticals related to muscle loss.

Although these findings support the potential of AJHW as a promising resource for the development of functional products and pharmaceuticals related to muscle loss, further studies are needed to validate its efficacy across different atrophy models, clarify the specific active compounds involved, and assess long-term safety.

## Figures and Tables

**Figure 1 ijms-26-03656-f001:**
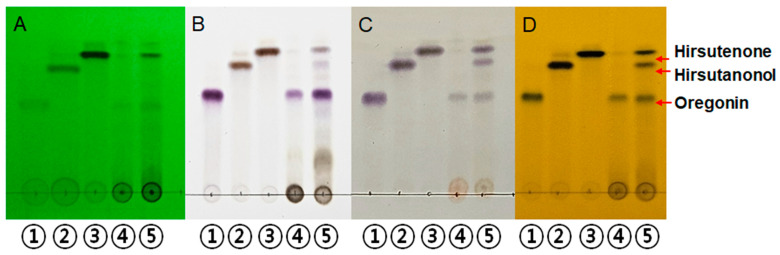
Thin Layer Chromatography. (**A**) UV 254 nm, (**B**) 10% H_2_SO_4_, (**C**) ρ-Anisaldehyde H_2_SO_4_, (**D**) FeCl_3_. The eluent system used was chloroform/methanol/water = 70:30:4 (*v*/*v*/*v*). ① oregonin standard, ② hirsutanonol standard, ③ hirsutenone standard, ④ *Alnus japonica* 50% EtOH extract, ⑤ *Alnus japonica* hot water extract (AJHW).

**Figure 2 ijms-26-03656-f002:**
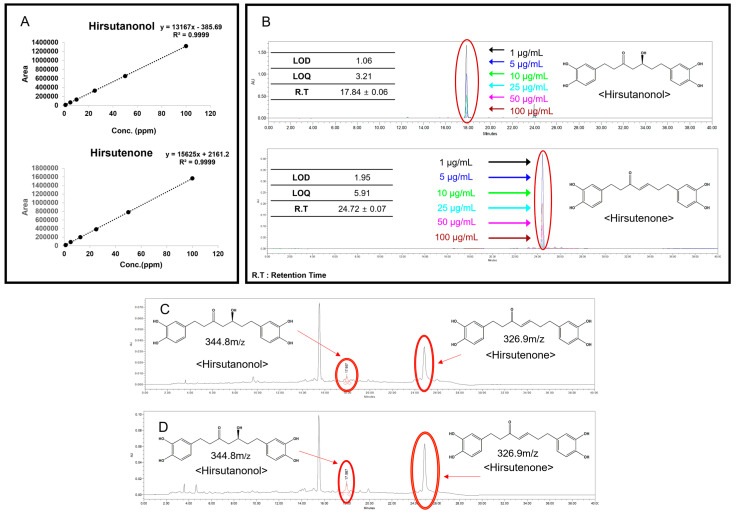
(**A**) Calibration curve and equation for hirsutanonol and hirsutenone, hirsutanonol: Y = 13,167X − 385.69 (R^2^ = 0.9999), hirsutenone: Y = 15,625X + 2161.2 (R^2^ = 0.9999); (**B**) HPLC chromatogram of hirsutanonol and hirsutenone; (**C**) HPLC chromatogram of *Alnus japonica* 50% EtOH extract at 1000 μg/mL; (**D**) HPLC chromatogram of AJHW at 1000 μg/mL.

**Figure 3 ijms-26-03656-f003:**
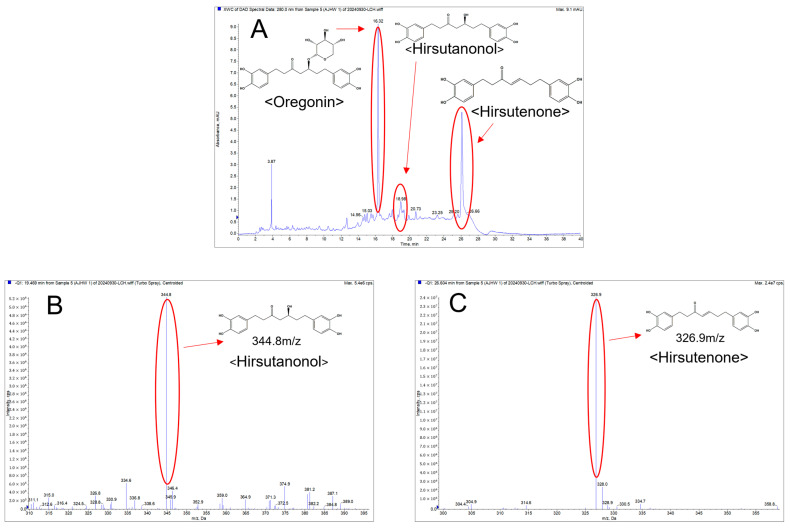
Negative mode LC-MS/MS of AJHW: (**A**) extracted ion chromatogram of AJHW, (**B**) total ion chromatogram, where the highlighted section corresponds to the mass value of hirsutanonol, and (**C**) total ion chromatogram, where the highlighted section corresponds to the mass value of hirsutenone.

**Figure 4 ijms-26-03656-f004:**
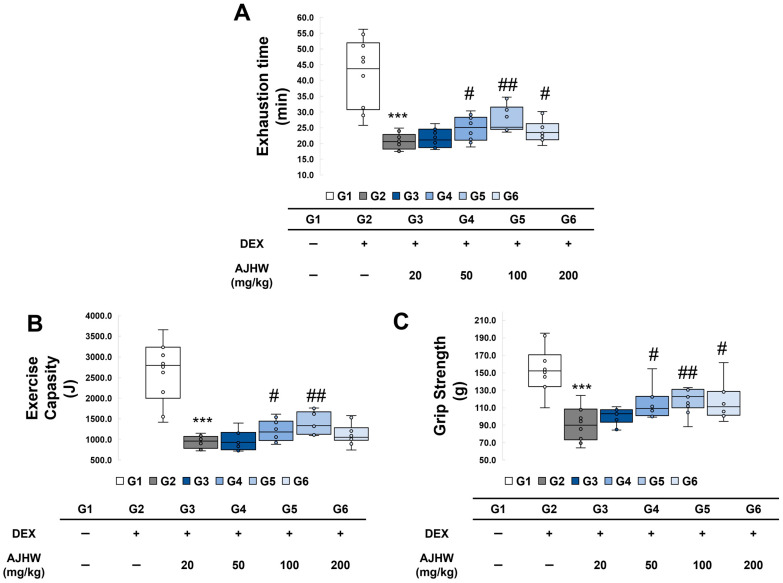
Effects of AJHW administration on exercise capacity and grip strength in mice with dexamethasone-induced muscle atrophy. (**A**) Endurance time to exhaustion, (**B**) exercise capacity, and (**C**) grip strength. All mice had an initial age of 8 weeks. G2–G6 were treated daily with dexamethasone (5 mg/kg) for 2 weeks. AJHW was administered daily for 4 weeks at 20 mg/kg (G3), 50 mg/kg (G4), 100 mg/kg (G5), and 200 mg/kg (G6). A total of ten animals was utilized in each group. Data are expressed as the mean ± SEM. *** *p* < 0.001 is significantly different from that of the G1 group. (G2). # *p* < 0.05, ## *p* < 0.01, are significantly different from that of G2 group. (G3, G4, G5, G6).

**Figure 5 ijms-26-03656-f005:**
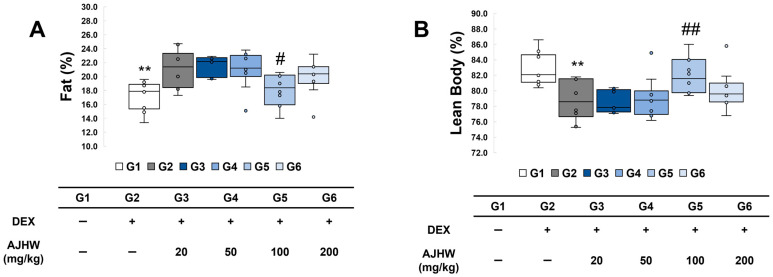
Effects of AJHW administration on fat and lean body percentage in mice with dexamethasone-induced muscle atrophy. (**A**) Fat percentage, (**B**) lean body percentage. All mice had an initial age of 8 weeks. G2–G6 were treated daily with dexamethasone (5 mg/kg) for 2 weeks. AJHW was administered daily for 4 weeks at 20 mg/kg (G3), 50 mg/kg (G4), 100 mg/kg (G5), and 200 mg/kg (G6). A total of ten animals was utilized in each group. Values are expressed as the mean ± SEM. ** *p* < 0.01 is significantly different from that of the G1 group. (G2). # *p* < 0.05, ## *p* < 0.01 are significantly different from that of G2 group. (G3, G4, G5, G6).

**Figure 6 ijms-26-03656-f006:**
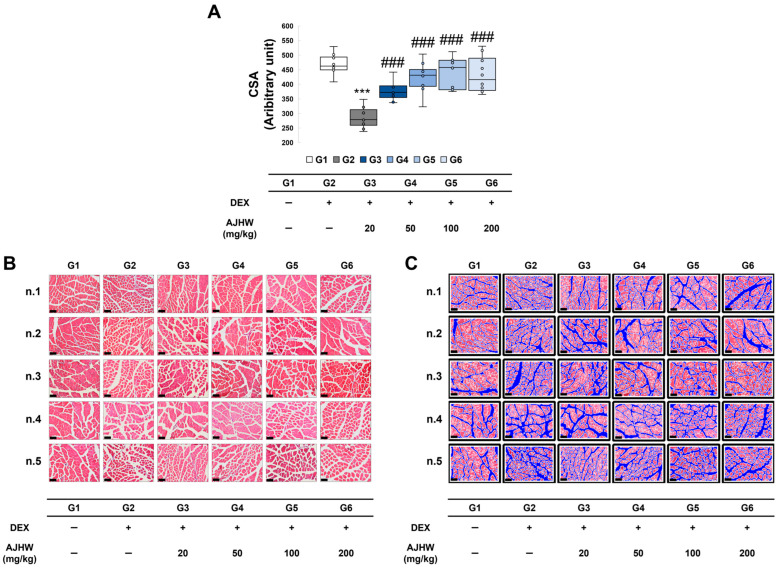
Effect of AJHW administration on skeletal muscle atrophy in mice with dexamethasone-induced muscle atrophy. (**A**) Quantitative data on the cross-sectional area of myofibers in the tibialis anterior (TA) muscle. (**B**) Representative images of the tibialis anterior (TA) muscle stained with H&E. (**C**) An image highlighting the muscle fiber area by coloring the surrounding region in blue. Only the red-stained muscle fibers were used for CSA measurement, utilizing ImageJ software (Version 1.54h). For CSA analysis (**A**), we applied the minimal Feret’s diameter method for calculation and quantified the results using ImageJ software. The scale bar is 100 μm. All mice had an initial age of 8 weeks. G2–G6 were treated daily with dexamethasone (5 mg/kg) for 2 weeks. AJHW was administered daily for 4 weeks at 20 mg/kg (G3), 50 mg/kg (G4), 100 mg/kg (G5), and 200 mg/kg (G6). A total of ten animals was utilized in each group. Values are presented as the mean ± SEM. *** *p* < 0.001 is significantly different from that of the G1 group. (G2). ### *p* < 0.001 is significantly different from that of the G2 group. (G3, G4, G5, G6).

**Figure 7 ijms-26-03656-f007:**
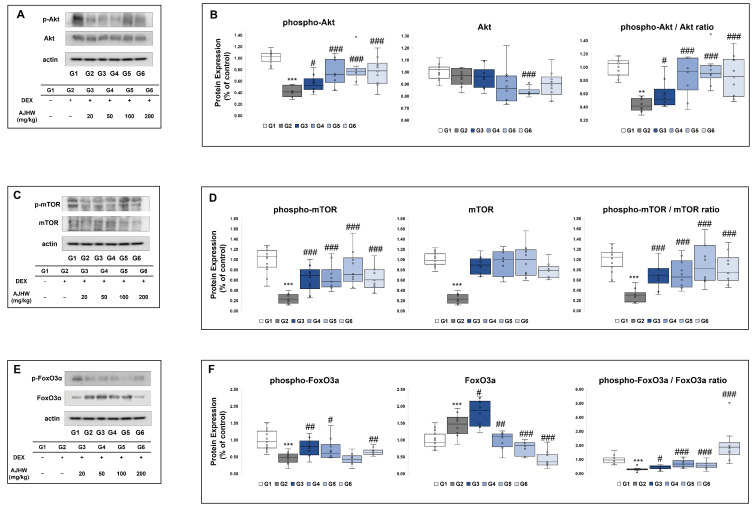
The effect of AJHW administration on the Akt/mTOR and FoxO3α signaling pathways in the gastrocnemius muscle of dexamethasone-induced muscle atrophic mice was investigated. Western blot analysis was used to analyze the protein expression levels of (**A**) phospho-Akt and Akt, (**C**) phospho-mTOR and mTOR, and (**E**) phospho-FoxO3α and FoxO3α. (**B**,**D**,**F**) A quantitative analysis of the western blot results was performed. Each protein expression level was normalized to that of actin and expressed as a relative value to the G1 group. All mice had an initial age of 8 weeks. G2–G6 were treated daily with dexamethasone (5 mg/kg) for 2 weeks. AJHW was administered daily for 4 weeks at 20 mg/kg (G3), 50 mg/kg (G4), 100 mg/kg (G5), and 200 mg/kg (G6). A total of ten animals was utilized in each group. Values are expressed as the mean ± SEM. ** *p* < 0.01, *** *p* < 0.001 are significantly different from that of G1 group. (G2). # *p* < 0.05, ## *p* < 0.01, ### *p* < 0.001 are significantly different from that of G2 group. (G3, G4, G5, G6).

**Figure 8 ijms-26-03656-f008:**
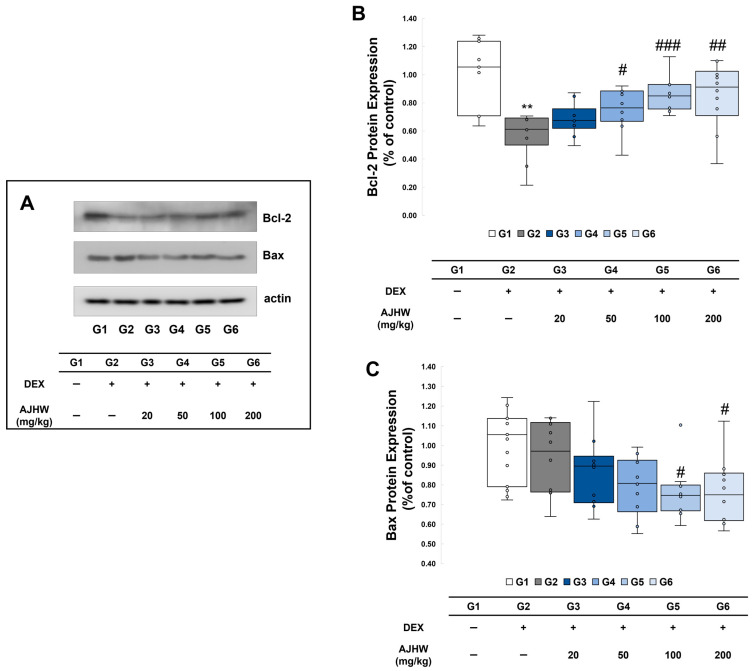
Effect of AJHW administration on the expression of apoptosis regulatory proteins. (**A**) Western blot analysis was used to analyze the protein expressions of Bcl-2 and Bax, (**B**) specifically Bcl-2, and (**C**) Bax. Quantitative analysis of the western blot results was performed on Bcl-2 (**B**) and Bax (**C**). Each protein expression level was normalized to that of actin and expressed relative to the G1 group. All mice had an initial age of 8 weeks. G2–G6 were treated daily with dexamethasone (5 mg/kg) for 2 weeks. AJHW was administered daily for 4 weeks at 20 mg/kg (G3), 50 mg/kg (G4), 100 mg/kg (G5), and 200 mg/kg (G6). A total of ten animals was utilized in each group. Values are stated as the mean ± SEM. ** *p* < 0.01 is significantly different from that of the G1 group. (G2). # *p* < 0.05, ## *p* < 0.01, ### *p* < 0.001 are significantly different from that of the G2 group. (G3, G4, G5, G6).

**Figure 9 ijms-26-03656-f009:**
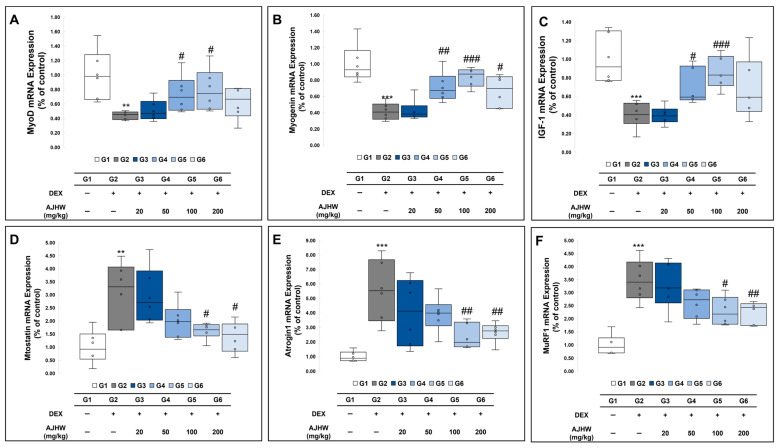
Effect of AJHW administration on mRNA expression levels for muscle synthesis and degradation-related genes in the soleus muscle (SOL) of dexamethasone-induced muscle atrophic mice. The relative mRNA expression levels of (**A**) MyoD, (**B**) myogenin, (**C**) IGF-1, (**D**) myostatin, (**E**) Atrogin1, and (**F**) MuRF1 were analyzed using real-time PCR. Target mRNA expression was normalized to that of GAPDH. All mice had an initial age of 8 weeks. G2–G6 were treated daily with dexamethasone (5 mg/kg) for 2 weeks. AJHW was administered daily for 4 weeks at 20 mg/kg (G3), 50 mg/kg (G4), 100 mg/kg (G5), and 200 mg/kg (G6). A total of ten animals was utilized in each group. Values are expressed as mean ± SEM. ** *p* < 0.01, *** *p* < 0.001 are significantly different from that of the G1 group. (G2). # *p* < 0.05, ## *p* < 0.01, ### *p* < 0.001 are significantly different from that of the G2 group. (G3, G4, G5, G6).

**Figure 10 ijms-26-03656-f010:**
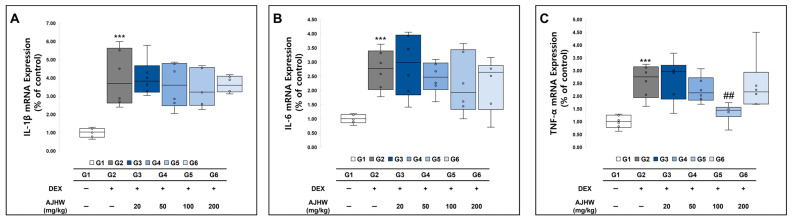
Effect of AJHW administration on mRNA expression of pro-inflammatory cytokines in the soleus muscle of dexamethasone-induced muscle atrophy mice. The relative mRNA expression levels of (**A**) IL-1β, (**B**) IL-6, and (**C**) TNF-α were analyzed using real-time PCR. Target mRNA expression was normalized to that of GAPDH. All mice had an initial age of 8 weeks. G2–G6 were treated daily with dexamethasone (5 mg/kg) for 2 weeks. AJHW was administered daily for 4 weeks at 20 mg/kg (G3), 50 mg/kg (G4), 100 mg/kg (G5), and 200 mg/kg (G6). A total of ten animals was utilized in each group. Values are expressed as mean ± SEM. *** *p* < 0.001 is significantly different from that of the G1 group. (G2). ## *p* < 0.01 is significantly different from that of the G2 group. (G3, G4, G5, G6).

**Figure 11 ijms-26-03656-f011:**
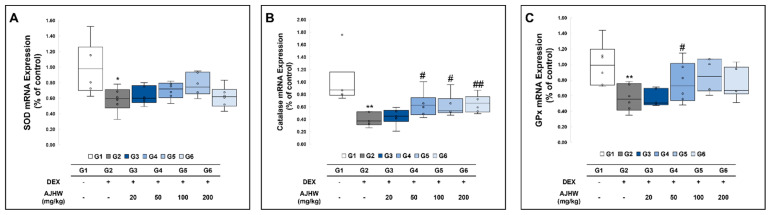
Effect of AJHW administration on the mRNA expression of antioxidant enzymes in the soleus muscle of dexamethasone-induced muscle atrophic mice. The relative mRNA expression levels of (**A**) SOD2, (**B**) catalase, and (**C**) GPx1 were analyzed by real-time PCR. Target mRNA expression was normalized to that of GAPDH. All mice had an initial age of 8 weeks. G2–G6 were treated daily with dexamethasone (5 mg/kg) for 2 weeks. AJHW was administered daily for 4 weeks at 20 mg/kg (G3), 50 mg/kg (G4), 100 mg/kg (G5), and 200 mg/kg (G6). A total of ten animals was utilized in each group. Values are reported as the mean ± SEM. * *p* < 0.05, ** *p* < 0.01 are significantly different from that of G1 group. (G2). # *p* < 0.05, ## *p* < 0.01 are significantly different from that of G2 group. (G3, G4, G5, G6).

**Figure 12 ijms-26-03656-f012:**
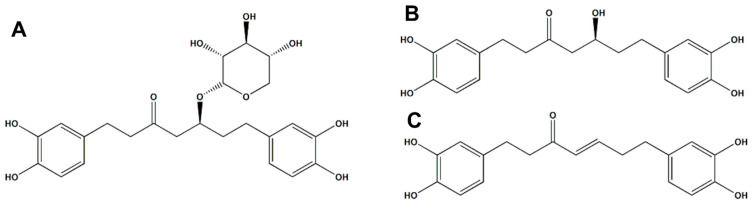
(**A**) The structure of oregonin, (**B**) the structure of hirsutanonol, (**C**) the structure of hirsutenone. The chemical structure was drawn using ChemDraw Ultra 7.0 (CambridgeSoft, Cambridge, MA, USA).

**Figure 13 ijms-26-03656-f013:**
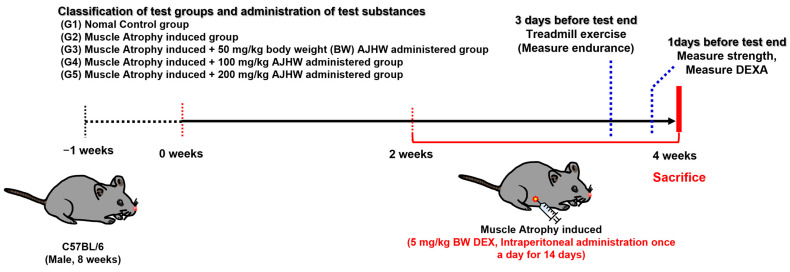
Experimental design and treatment.

**Table 1 ijms-26-03656-t001:** Body weight (BW).

	0 Week	1 Week	2 Week	3 Week	4 Week
G1	25.1 ± 0.4	27.5 ± 0.3	28.1 ± 0.4	29.8 ± 0.6	30.2 ± 0.5
G2	25.1 ± 0.3	27.1 ± 0.2	28.1 ± 0.2	27.6 ± 0.3 **	27.1 ± 0.4 ***
G3	25.2 ± 0.3	27.2 ± 0.4	28.0 ± 0.3	27.0 ± 0.5	27.5 ± 0.7
G4	25.2 ± 0.3	27.5 ± 0.6	27.7 ± 0.6	27.9 ± 0.8	27.8 ± 0.6
G5	25.1 ± 0.2	27.2 ± 0.4	28.3 ± 0.4	26.8 ± 0.6	28.5 ± 0.7
G6	25.1 ± 0.2	26.8 ± 0.3	27.9 ± 0.4	26.6 ± 0.5	27.3 ± 0.5

All mice had an initial age of 8 weeks. G2–G6 were treated daily with dexamethasone (5 mg/kg) for 2 weeks. AJHW was administered daily for 4 weeks at 20 mg/kg (G3), 50 mg/kg (G4), 100 mg/kg (G5), and 200 mg/kg (G6). A total of ten animals was utilized in each group. Values are provided as mean ± SEM. ** *p* < 0.01, *** *p* < 0.001 are significantly different from the G1 group. (G2).

**Table 2 ijms-26-03656-t002:** Muscle weight (g) and relative muscle weight (g/100 g body weight).

		G1	G2	G3	G4	G5	G6
Muscle weight (g)	QF	0.390 ± 0.015	0.305 ± 0.008 ***	0.313 ± 0.013	0.338 ± 0.010 #	0.365 ± 0.020 #	0.333 ± 0.015
	GA	0.325 ± 0.007	0.268 ± 0.006 ***	0.273 ± 0.008	0.285 ± 0.008	0.298 ± 0.010 #	0.286 ± 0.005 #
	SOL	0.0202 ± 0.0003	0.0171 ± 0.0002 ***	0.0174 ± 0.0008	0.0176 ± 0.0005	0.0189 ± 0.0004 ##	0.0176 ± 0.0004
	EDL	0.026 ± 0.002	0.019 ± 0.001 **	0.018 ± 0.001	0.021 ± 0.001	0.022 ± 0.001 #	0.020 ± 0.001
	TA	0.126 ± 0.003	0.102 ± 0.005 **	0.105 ± 0.011	0.111 ± 0.003	0.115 ± 0.006	0.110 ± 0.003
Relative muscle weight (g/100 g BW)	QF	1.295 ± 0.048	1.126 ± 0.027 **	1.144 ± 0.055	1.225 ± 0.045	1.279 ± 0.065 #	1.223 ± 0.060
	GA	1.079 ± 0.030	0.990 ± 0.025 *	0.997 ± 0.038	1.032 ± 0.039	1.053 ± 0.050	1.049 ± 0.017
	SOL	0.067 ± 0.001	0.063 ± 0.001 *	0.063 ± 0.002	0.063 ± 0.002	0.066 ± 0.001 #	0.065 ± 0.001
	EDL	0.088 ± 0.006	0.069 ± 0.004 *	0.066 ± 0.003	0.075 ± 0.003	0.076 ± 0.003	0.075 ± 0.003
	TA	0.418 ± 0.013	0.376 ± 0.017	0.389 ± 0.044	0.402 ± 0.015	0.405 ± 0.020	0.404 ± 0.015

All mice had an initial age of 8 weeks. G2–G6 were treated daily with dexamethasone (5 mg/kg) for 2 weeks. AJHW was administered daily for 4 weeks at 20 mg/kg (G3), 50 mg/kg (G4), 100 mg/kg (G5), and 200 mg/kg (G6). A total of ten animals was utilized in each group. Values are expressed as the mean ± SEM. * *p* < 0.05, ** *p* < 0.01, *** *p* < 0.001 are significantly different from that of G1 group. (G2). # *p* < 0.05, ## *p* < 0.01 are significantly different from that of G2 group. (G3, G4, G5, G6).

**Table 3 ijms-26-03656-t003:** Changes in glutathione content and antioxidant enzyme activity in serum.

	G1	G2	G3	G4	G5	G6
Glutathione (μM)	12.43 ± 2.57	4.38 ± 0.75 **	8.25 ± 0.68 ##	7.47 ± 0.78 #	8.59 ± 0.68 ###	8.71 ± 0.87 ##
SOD (U/mL)	3.59 ± 0.23	2.33 ± 0.28 *	1.96 ± 0.22	2.63 ± 0.17	2.68 ± 0.19	3.68 ± 0.25 #
Catalase (nM/min/mL)	41.0 ± 4.0	26.1 ± 3.6 *	31.9 ± 3.0	32.9 ± 3.2	37.3 ± 3.7 #	36.7 ± 5.0
GPx (nM/min/mL)	198.7 ± 5.0	138.9 ± 5.4 **	159.1 ± 13.9	175.1 ± 10.4 ##	189.1 ± 6.2 ###	189.2 ± 6.9 ###

All mice had an initial age of 8 weeks. G2–G6 were treated daily with dexamethasone (5 mg/kg) for 2 weeks. AJHW was administered daily for 4 weeks at 20 mg/kg (G3), 50 mg/kg (G4), 100 mg/kg (G5), and 200 mg/kg (G6). A total of ten animals was utilized in each group. Values are expressed as the mean ± SEM. * *p* < 0.05, ** *p* < 0.01 are significantly different from that of G1 group. (G2). # *p* < 0.05, ## *p* < 0.01, ### *p* < 0.001 are significantly different from that of G2 group. (G3, G4, G5, G6).

**Table 4 ijms-26-03656-t004:** Information regarding the antibodies used in the experiments.

Antibodies	Details	Manufacturer
Antibody: phospho-mTOR (Ser253)	#5536	Cell Signaling Technology (Danvers, MA, USA)
Antibody: mTOR	#2972	Cell Signaling Technology
Antibody: phospho-FoxO3α (Ser253)	#9466	Cell Signaling Technology
Antibody: FoxO3α	#2497	Cell Signaling Technology
Antibody: phospho-Akt (Ser)	#4051	Cell Signaling Technology
Antibody: Akt	#9272	Cell Signaling Technology
Antibody: Bax	#2772	Cell Signaling Technology
Antibody: Bcl-2	#3498	Cell Signaling Technology
Antibody: Actin	#3700	Cell Signaling Technology

**Table 5 ijms-26-03656-t005:** Primer sequences for the real-time PCR used in this study.

mRNA	Primer Sequences	
Atrogin-1	Forward	5′-GCCCTCCACACTAGTTGACC-3′
	Reverse	5‘-GACGGATTGACAGCCAGGAA-3′
Catalase	Forward	5′-GAACGAGGAGGAGAGGAAAC-3′
	Reverse	5′-TGAAATTCTTGACCGCTTTC-3′
GPx1	Forward	5′-CAGGTCGGACGTACTTGAG-3′
	Reverse	5′-CAGGTCGGACGTACTTGAG-3′
IGF-1	Forward	5′-GTGGATGCTCTTCAGTTCGTGTG-3′
	Reverse	5′-TCCAGTCTCCTCAGATCACAGC-3′
IL-1β	Forward	5′-TGGACCTTCCAGGATGAGGACA-3′
	Reverse	5′-GTTCATCTCGGAGCCTGTAGTG-3′
IL-6	Forward	5′-CCTCTGGTCTTCTGGAGTACC-3′
	Reverse	5′-ACTCCTTCTGTGACTCCAGC-3′
MuRF1	Forward	5′-GAGGGCCATTGACTTTGGGA-3′
	Reverse	5′-TTTACCCTCTGTGGTCACGC-3′
MyoD1	Forward	5′-GCACTACAGTGGCGACTCAGAT-3′
	Reverse	5′-TAGTAGGCGGTGTCGTAGCCAT-3′
Myogenin	Forward	5′-CCATCCAGTACATTGAGCGCCT-3′
	Reverse	5′-CTGTGGGAGTTGCATTCACTGG-3′
Myostatin	Forward	5′-ACTGGACCTCTCGATAGAACACTC-3′
	Reverse	5′-ACTTAGTGCTGTGTGTGTGGAGAT-3′
SOD2	Forward	5′-ATCAGGACCCATTGCAAGGA-3′
	Reverse	5′-AGGTTTCACTTCTTGCAAGCT-3′
TNF-α	Forward	5′-GGTGCCTATGTCTCAGCCTCTT-3′
	Reverse	5′-GCCATAGAACTGATGAGAGGGAG-3′
GAPDH	Forward	5′-TGGGTGTGAACCATGAGAAG-3′
	Reverse	5′-GCTAAGCAGTTGGTGGTGC-3′

## Data Availability

The original contributions presented in the study are included in the article; further inquiries can be directed to the corresponding author.

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
