# Peer review of "Effects of *Alnus japonica* Pilot Scale Hot Water Extracts on a Model of Dexamethasone-Induced Muscle Loss and Muscle Atrophy in C57BL/6 Mice"

_ijms, 2025, doi:10.3390/ijms26083656_

Round 1

Reviewer 1 Report (New Reviewer)

Comments and Suggestions for Authors

After carefully review the article ‘ The impact of Alnus japonica Pilot scale Hot Water Extracts on a model of Sarcopenia-induced C57BL/6 mice’. While the authors to attempt to examine the therapeutic effects of Hot Water Extracts, the study is not complete and lacks scientific rigor.

Although the plots all showed significant restoration of atrophy which was induced by Dex after AJHW, there is a flaw in this study that the authors did not prove if the AJHW directly worked on Dex not the atrophy. The authors should use real sarcopenia models (aging or transgenic). and test the AJHW as well. Or they could use a cellular model to examine AJHW. For example, C2C12 cells.

Author Response

Reviewer 1

Comments 1: 

After carefully review the article ‘The impact of Alnus japonica Pilot scale Hot Water Extracts on a model of Sarcopenia-induced C57BL/6 mice’. While the authors to attempt to examine the therapeutic effects of Hot Water Extracts, the study is not complete and lacks scientific rigor.

Although the plots all showed significant restoration of atrophy which was induced by Dex after AJHW, there is a flaw in this study that the authors did not prove if the AJHW directly worked on Dex not the atrophy. The authors should use real sarcopenia models (aging or transgenic). and test the AJHW as well. Or they could use a cellular model to examine AJHW. For example, C2C12 cells.

Response 1:
We sincerely appreciate your valuable feedback and insightful comments, which have provided critical perspectives to further enhance our study.

Regarding your concern, our research team has previously conducted an in vitro study using C2C12 cells, as suggested. This study was published last year [1] and established the basis for our current in vivo investigation. As described in the manuscript, our primary goal is to develop AJHW as a potential natural pharmaceutical or functional health product. To achieve this, we have conducted both in vitro mechanism studies and in vivo efficacy evaluation.

Furthermore, we are currently planning clinical trials based on our research findings. Our study extends beyond lab-scale experiments, aiming to contribute substantively to academia, industry, and public healthcare, ultimately facilitating commercialization.

We acknowledge the significance of your recommendation concerning the use of a sarcopenia model for further validation. In future research, we will integrate this suggestion by conducting cross-validation using appropriate and established sarcopenia models.

Once again, we greatly appreciate your constructive feedback and thoughtful suggestions.

  1. An, D.H.; Lee, C.H.; Kwon, Y.; Kim, T.H.; Kim, E.J.; Jung, J.I.; Min, S.; Cheong, E.J.; Kim, S.; Kim, H.K.; et al. Effects of Alnus japonica Hot Water Extract and Oregonin on Muscle Loss and Muscle Atrophy in C2C12 Murine Skeletal Muscle Cells. Pharmaceuticals 2024, 17, doi:10.3390/ph17121661.

Reviewer 2 Report (New Reviewer)

Comments and Suggestions for Authors

The manuscript entitled “The Impact of Alnus japonica Pilot Scale Hot Water Extracts on a Model of Sarcopenia-induced C57BL/6 Mice” by Dy Jung et al., described a new method to extract the oregonin using hot water instead of ethanol. Oregonin has been described as an antioxidant, anti-inflammatory, antitumor, anti-bacterial, and hepatoprotective compound from Alnus japonica. The Authors test the dose-dependent effect of the extract in a murine model of sarcopenia. After demonstrating the evidence of the presence of the compound in the extract, the Authors measured mechanical and biochemical parameters in a dexamethasone-dependent sarcopenia model. The manuscript is well-written, clear, and well-presented and it is an interesting study. However, there are three major points and a few minor concerns that the Authors should correct before a final decision is made.

Major Points:

1)    The Authors use young adult mice for a sarcopenia study. To simulate the characteristics of sarcopenia, the Authors injected dexamethasone via i.p. twice. While dexamethasone can be used to exacerbate the effect of sarcopenia, it does not induce it in young adult mice. The effects observed in the manuscript with the injection of dexamethasone is glucocorticoid-dependent muscle atrophy, which shares some of the diagnosis of sarcopenia. Sarcopenia is an aging-associated disease, and its diagnosis is not based solely on the effects caused by dexamethasone. Therefore, the Authors should adjust the title and content of the manuscript accordingly.

2)    The statistical analyses must be redone because the Authors used one-way ANOVA when there are two factors, namely dexamethasone and the presence or absence of the extract. Moreover, the Authors must describe it in the text of the Results section and present the F values.

3)    The content and activity measurements of glutathione, catalase, SOD, and GPx in plasma should have been done in muscle rather than in serum. The reason is these molecules and enzymes can be found in several tissues, such as erythrocytes, kidneys, and liver. Moreover, dexamethasone can induce tissue damage and the release of these molecules into the plasma.  

Minor points:

1)    All plots should be presented as box-plot instead of bars.

2)    In the Introduction, the Authors should clarify better the rationale for evaluating the content of Hirsutanonol and Hirsutenone in the study.

3)     The Authors should clarify the differences between groups 2-6 in the Results section, as well as the initial age of all groups.

4)    Please correct the typo, adding space between Figure and 4C.

5)    The country specification of the company should go after the company's name and not at the end of the sentence.

6)    The entire CSA analysis must be redone because the Authors should use the minimal Ferret's diameter correction.

7)    In Figure 6, the Authors should also include nuclei staining (for instance DAPI) to show whether there is or not the presence of muscle regeneration and staining of fat and fibrotic tissues.

8)    Please correct the text to Figure 13.

9)    How did the Authors had defined the dosage of the 5mg/ kg dexamethasone in mice?

10) The Authors should clarify the adaptation process for the animals to the treadmill and to the grip test. Without proper training on both devices, the performance of the mice is poor and therefore, the data is not reliable. 

Author Response

Reviewer 2

Comments 1: 

The Authors use young adult mice for a sarcopenia study. To simulate the characteristics of sarcopenia, the Authors injected dexamethasone via i.p. twice. While dexamethasone can be used to exacerbate the effect of sarcopenia, it does not induce it in young adult mice. The effects observed in the manuscript with the injection of dexamethasone is glucocorticoid-dependent muscle atrophy, which shares some of the diagnosis of sarcopenia. Sarcopenia is an aging-associated disease, and its diagnosis is not based solely on the effects caused by dexamethasone. Therefore, the Authors should adjust the title and content of the manuscript accordingly

Response 1:
We sincerely appreciate your thoughtful feedback. As you have correctly pointed out, muscle loss and atrophy alone do not comprehensively define sarcopenia. However, our research team considers these factors to be crucial components in sarcopenia-like conditions. Therefore, we applied dexamethasone treatment in a mouse model to induce muscle loss and atrophy.

To establish this model, we carefully referred to multiple studies that have employed dexamethasone treatment for inducing muscle atrophy [1-4]. Furthermore, in our previously published in vitro study on AJHW, we utilized the C2C12 model with dexamethasone treatment to construct a muscle atrophy model [5].

We have revised the manuscript title and content to reflect your suggestions. We greatly appreciate your insightful suggestions and your effort in pointing out this important aspect.

  1. Kim, J.W.; Ku, S.-K.; Han, M.H.; Kim, K.Y.; Kim, S.G.; Kim, G.-Y.; Hwang, H.J.; Kim, B.W.; Kim, C.M.; Choi, Y.H. The administration of Fructus Schisandrae attenuates dexamethasone-induced muscle atrophy in mice. Int J Mol Med 2015, 36, 29-42, doi:10.3892/ijmm.2015.2200.
  2. Lee, H.; Seon Lee, K.; Hye Jeong, J.; Soo Yoon, J.; Hwan Hwang, S.; Kim, S.-Y.; Hum Yeon, S.; Ryu, J.-H. Extract of Alnus japonica prevents dexamethasone-induced muscle atrophy in mice. Journal of Functional Foods 2023, 101, 105419, doi:https://doi.org/10.1016/j.jff.2023.105419.
  3. Kim, M.S.; Park, S.; Kwon, Y.; Kim, T.; Lee, C.H.; Jang, H.; Kim, E.J.; Jung, J.I.; Min, S.; Park, K.-H.; et al. Effects of Ulmus macrocarpa Extract and Catechin 7-O-β-D-apiofuranoside on Muscle Loss and Muscle Atrophy in C2C12 Murine Skeletal Muscle Cells. Current Issues in Molecular Biology 2024, 46, 8320-8339, doi:https://doi.org/10.3390/cimb46080491.
  4. Lee, C.H.; Kwon, Y.; Park, S.; Kim, T.; Kim, M.S.; Kim, E.J.; Jung, J.I.; Min, S.; Park, K.-H.; Jeong, J.H.; et al. The Impact of Ulmus macrocarpa Extracts on a Model of Sarcopenia-Induced C57BL/6 Mice. International Journal of Molecular Sciences 2024, 25, doi:10.3390/ijms25116197.
  5. An, D.H.; Lee, C.H.; Kwon, Y.; Kim, T.H.; Kim, E.J.; Jung, J.I.; Min, S.; Cheong, E.J.; Kim, S.; Kim, H.K.; et al. Effects of Alnus japonica Hot Water Extract and Oregonin on Muscle Loss and Muscle Atrophy in C2C12 Murine Skeletal Muscle Cells. Pharmaceuticals 2024, 17, doi:10.3390/ph17121661.

Comments 2: 

 The statistical analyses must be redone because the Authors used one-way ANOVA when there are two factors, namely dexamethasone and the presence or absence of the extract. Moreover, the Authors must describe it in the text of the Results section and present the F values.

Response 2:
We sincerely appreciate your valuable feedback. As you pointed out, we conducted group comparisons using one-way ANOVA. We acknowledge that, for analyzing multiple factors, an alternative approach such as two-way ANOVA would be more appropriate to ensure valid F values.

However, due to the limitations of our current experimental model, the available data are not sufficient for the application of two-way ANOVA. Our study was designed with reference to multiple prior studies [1-4], but as you correctly noted, we did not completely account for additional comparative methods for multiple factors. Additionally, given our experimental design, one-way ANOVA was considered a more appropriate method for comparing the groups in this study.

While further experiments would be necessary to address this limitation, unfortunately, a significant amount of time has passed since the completion of our study, making additional data collection difficult at this stage. Nonetheless, we recognize the importance of your suggestion and will consider it in the experimental design of our future research.

We would sincerely appreciate your understanding of this situation.

Once again, we sincerely appreciate your insightful and constructive feedback.

  1. Lee, H.; Seon Lee, K.; Hye Jeong, J.; Soo Yoon, J.; Hwan Hwang, S.; Kim, S.-Y.; Hum Yeon, S.; Ryu, J.-H. Extract of Alnus japonica prevents dexamethasone-induced muscle atrophy in mice. Journal of Functional Foods 2023, 101, 105419, doi:https://doi.org/10.1016/j.jff.2023.105419.
  2. Lee, D.-Y.; Chun, Y.-S.; Kim, J.-K.; Lee, J.-O.; Ku, S.-K.; Shim, S.-M. Curcumin Attenuates Sarcopenia in Chronic Forced Exercise Executed Aged Mice by Regulating Muscle Degradation and Protein Synthesis with Antioxidant and Anti-inflammatory Effects. Journal of Agricultural and Food Chemistry 2021, 69, 6214-6228, doi:10.1021/acs.jafc.1c00699.
  3. Kim, J.W.; Ku, S.-K.; Han, M.H.; Kim, K.Y.; Kim, S.G.; Kim, G.-Y.; Hwang, H.J.; Kim, B.W.; Kim, C.M.; Choi, Y.H. The administration of Fructus Schisandrae attenuates dexamethasone-induced muscle atrophy in mice. Int J Mol Med 2015, 36, 29-42, doi:10.3892/ijmm.2015.2200.
  4. Lee, C.H.; Kwon, Y.; Park, S.; Kim, T.; Kim, M.S.; Kim, E.J.; Jung, J.I.; Min, S.; Park, K.-H.; Jeong, J.H.; et al. The Impact of Ulmus macrocarpa Extracts on a Model of Sarcopenia-Induced C57BL/6 Mice. International Journal of Molecular Sciences 2024, 25, doi:10.3390/ijms25116197.

Comments 3: 

 The content and activity measurements of glutathione, catalase, SOD, and GPx in plasma should have been done in muscle rather than in serum. The reason is these molecules and enzymes can be found in several tissues, such as erythrocytes, kidneys, and liver. Moreover, dexamethasone can induce tissue damage and the release of these molecules into the plasma.  

Response 3:
Thank you for your insightful comment. Since dexamethasone treatment induces oxidative stress, we demonstrated the antioxidant effects of AJHW in Figure 8. Additionally, our previous in vitro study has already confirmed its associated antioxidant properties [1].

In this study, our analysis was not limited to muscle tissue but extended to antioxidant molecules and enzymes expressed in the blood of mice. This approach enabled us to assess dexamethasone-induced oxidative stress and the subsequent changes following AJHW treatment. As the evaluation of antioxidant activity is a crucial component in studies on sarcopenia-like conditions, this experiment was conducted.

We acknowledge the validity of your suggestion and recognize the importance of directly measuring these markers in muscle tissue. However, we would like to emphasize that our methodology aligns with widely accepted research approaches, as supported by multiple references [2-4]. We sincerely appreciate your constructive feedback and will take this aspect into consideration in our future studies.

  1. An, D.H.; Lee, C.H.; Kwon, Y.; Kim, T.H.; Kim, E.J.; Jung, J.I.; Min, S.; Cheong, E.J.; Kim, S.; Kim, H.K.; et al. Effects of Alnus japonica Hot Water Extract and Oregonin on Muscle Loss and Muscle Atrophy in C2C12 Murine Skeletal Muscle Cells. Pharmaceuticals 2024, 17, doi:10.3390/ph17121661.
  2. Lee, D.-Y.; Chun, Y.-S.; Kim, J.-K.; Lee, J.-O.; Ku, S.-K.; Shim, S.-M. Curcumin Attenuates Sarcopenia in Chronic Forced Exercise Executed Aged Mice by Regulating Muscle Degradation and Protein Synthesis with Antioxidant and Anti-inflammatory Effects. Journal of Agricultural and Food Chemistry 2021, 69, 6214-6228, doi:10.1021/acs.jafc.1c00699.
  3. Oh, H.-J.; Jin, H.; Lee, B.-Y. The non-saponin fraction of Korean Red Ginseng ameliorates sarcopenia by regulating immune homeostasis in 22–26-month-old C57BL/6J mice. Journal of Ginseng Research 2022, 46, 809-818, doi:https://doi.org/10.1016/j.jgr.2022.05.007.
  4. Bellanti, F.; Lo Buglio, A.; Quiete, S.; Dobrakowski, M.; Kasperczyk, A.; Kasperczyk, S.; Vendemiale, G. Sarcopenia Is Associated with Changes in Circulating Markers of Antioxidant/Oxidant Balance and Innate Immune Response. Antioxidants 2023, 12, doi:10.3390/antiox12111992.

Comments:  Minor

1) All plots should be presented as box-plot instead of bars.

Minor Response 1:
Thank you for your valuable suggestion. In response to your feedback, we have revised all plots to be presented in box-plot format.

2) In the Introduction, the Authors should clarify better the rationale for evaluating the content of Hirsutanonol and Hirsutenone in the study.

Minor Response 2:
Thank you for your valuable feedback. Oregonin is a well-characterized bioactive compound derived from Alnus species. Among its two aglycone derivatives, Hirsutanonol is produced through the hydrolysis of Oregonin, resulting in the removal of the xylose moiety. Hirsutenone, in contrast, is formed from Hirsutanonol via a dehydration reaction [1-5].

Previous in vitro studies have primarily focused on Oregonin, whereas investigations into Hirsutanonol and Hirsutenone remain limited. Given their potential biological significance, further research is warranted to elucidate their role in muscle atrophy and to enhance our understanding of Alnus extracts. Therefore, this study conducted a phytochemical analysis to quantify the content of Hirsutanonol and Hirsutenone alongside Oregonin.

In response to your feedback, we have revised the Introduction section to more clearly highlight the significance of evaluating these compounds.

We appreciate your insightful suggestion.

  1. Choi, S.E. Chemotaxonomic Significance of Oregonin in Alnus Species. Asian Journal of Chemistry 2013, 25, 6989-6990, doi:10.14233/ajchem.2013.15090
  2. Choi, S.E.; Park, K.H.; Jeong, M.S.; Kim, H.H.; Lee, D.I.; Joo, S.S.; Lee, C.S.; Bang, H.; Choi, Y.W.; Lee, M.-K.; et al. Effect of Alnus japonica extract on a model of atopic dermatitis in NC/Nga mice. Journal of Ethnopharmacology 2011, 136, 406-413, doi:https://doi.org/10.1016/j.jep.2010.12.024.
  3. Choi, S.E. Extraction method and physiological activity of high content oregonin derived from plant of Alnus sibirica Fisch. ex Turcz. Korean Journal of Pharmacognosy 2019, 50, 165-174.
  4. Choi, S.E.; Kim, K.H.; Kwon, J.H.; Kim, S.B.; Kim, H.W.; Lee, M.W. Cytotoxic activities of diarylheptanoids from Alnus japonica. Archives of Pharmacal Research 2008, 31, 1287-1289, doi:10.1007/s12272-001-2108-z.
  5. Sun Eun, C.; Kwan Hee, P.; Manh Heun, K.; Jeong Hwa, S.; Hye Young, J.; Min Won, L. Diarylheptanoids from the Bark of Alnus pendula Matsumura. Natural Product Sciences 2012, 18, 106-110.

3) The Authors should clarify the differences between groups 2-6 in the Results section, as well as the initial age of all groups.

Minor Response 3:
We have revised the legend to include this information, making the distinctions between groups 2–6 clearer. Additionally, we have clarified the initial age of all groups in the Results section.

4) Please correct the typo, adding space between Figure and 4C.

Minor Response 4:
The typo has been corrected as requested. Thank you for your attention to detail.

5) The country specification of the company should go after the company's name and not at the end of the sentence.

Minor Response 5:
Thank you for your meticulous review. We have corrected this accordingly.

6) The entire CSA analysis must be redone because the Authors should use the minimal Ferret's diameter correction.

Minor Response 6:

Thank you for your insightful comment. We quantified the CSA using ImageJ software and acknowledge the importance of incorporating the minimal Ferret’s diameter correction. In response to your feedback, we have added an additional figure to provide this information. CSA analysis was conducted by recognizing only the red-stained muscle fibers. We appreciate your valuable suggestion.

7) In Figure 6, the Authors should also include nuclei staining (for instance DAPI) to show whether there is or not the presence of muscle regeneration and staining of fat and fibrotic tissues.

Minor Response 7:

We acknowledge your valuable feedback and recognize the importance of including nuclei staining to assess muscle regeneration as well as fat and fibrotic tissue staining. However, due to the time constraints of the revision period, it is unfortunately not feasible to conduct additional animal experiments under the same conditions within the given timeframe. We truly appreciate your insightful suggestion, and we regret that we are unable to incorporate this aspect in the current study. However, we will take your recommendation into account and ensure that nuclei staining is included in the experimental design of our future research. Thank you again for your constructive input.

8) Please correct the text to Figure 13.

Minor Response 8:

The text for Figure 13 has been corrected as requested. Thank you for your attention to detail

9) How did the Authors had defined the dosage of the 5mg/ kg dexamethasone in mice?

Minor Response 9:
Thank you for highlighting this important aspect. The dosage and duration of dexamethasone administration for inducing muscle atrophy have been reported with variation in previous studies [1-5]. In our prior study, we determined that administering 5 mg/kg of dexamethasone for 14 days resulted in a significant induction of muscle atrophy [6]. Based on this established protocol, we adopted the same dosage in the present study. We appreciate your valuable feedback.

  1. Yoshioka, Y.; Kubota, Y.; Samukawa, Y.; Yamashita, Y.; Ashida, H. Glabridin inhibits dexamethasone-induced muscle atrophy. Archives of Biochemistry and Biophysics 2019, 664, 157-166, doi:https://doi.org/10.1016/j.abb.2019.02.006.
  2. Shen, S.; Liao, Q.; Liu, J.; Pan, R.; Lee, S.M.-Y.; Lin, L. Myricanol rescues dexamethasone-induced muscle dysfunction via a sirtuin 1-dependent mechanism. Journal of Cachexia, Sarcopenia and Muscle 2019, 10, 429-444, doi:https://doi.org/10.1002/jcsm.12393.
  3. Oh, S.; Choi, C.H.; Lee, B.-J.; Park, J.-H.; Son, K.-H.; Byun, K. Fermented Oyster Extract Attenuated Dexamethasone-Induced Muscle Atrophy by Decreasing Oxidative Stress. Molecules 2021, 26, doi:10.3390/molecules26237128.
  4. Lee, M.-K.; Choi, J.-W.; Choi, Y.H.; Nam, T.-J. Pyropia yezoensis Protein Supplementation Prevents Dexamethasone-Induced Muscle Atrophy in C57BL/6 Mice. Marine Drugs 2018, 16, doi:10.3390/md16090328.
  5. Lee, H.; Seon Lee, K.; Hye Jeong, J.; Soo Yoon, J.; Hwan Hwang, S.; Kim, S.-Y.; Hum Yeon, S.; Ryu, J.-H. Extract of Alnus japonica prevents dexamethasone-induced muscle atrophy in mice. Journal of Functional Foods 2023, 101, 105419, doi:https://doi.org/10.1016/j.jff.2023.105419.
  6. Lee, C.H.; Kwon, Y.; Park, S.; Kim, T.; Kim, M.S.; Kim, E.J.; Jung, J.I.; Min, S.; Park, K.-H.; Jeong, J.H.; et al. The Impact of Ulmus macrocarpa Extracts on a Model of Sarcopenia-Induced C57BL/6 Mice. International Journal of Molecular Sciences 2024, 25, doi:10.3390/ijms25116197.

10) The Authors should clarify the adaptation process for the animals to the treadmill and to the grip test. Without proper training on both devices, the performance of the mice is poor and therefore, the data is not reliable. 

Minor Response 10:
Thank you for your insightful comment. We have revised the manuscript to include details on the adaptation process for the animals to the treadmill and grip test to ensure clarity and reliability of the data. We appreciate your valuable feedback.

Round 2

Reviewer 2 Report (New Reviewer)

Comments and Suggestions for Authors

The authors of the revised manuscript with the adapted title “Effects of Alnus japonica pilot scale hot water extracts on a model of dexamethasone-induced muscle loss and muscle atrophy in C57BL/6 mice” by Du Jang et al., have positively accept most of the suggested comments, correcting all typos, and thereby improving the quality of the manuscript. However, there are three key points that the Authors did not address in the revised version of the manuscript.

One is that the text does not explicitly mention that the CSA was analyzed using the minimal Feret’s diameter. This detail is mentioned in the rebuttal letter but not in the manuscript. Since the Authors changed the data representation from bars plot to box-plot as requested by the Reviewer, it is unclear whether they actually recalculated CSA using the minimal Feret's diameter technique or simply changed the data representation.

The second is that the Authors continue to assume that the antioxidant enzymes activity values found in the serum reflect those in muscle tissue. However, without proper experimental validation, this remains mere speculation. The mentioned enzymes can also be found in erythrocytes, which have high expression of these enzymes, or from other than muscle tissues. The Authors should clarify this in the discussion and acknowledge the potential sources of these enzymes.

The third point is, again, the statistical analysis. The Authors persist in using the one-way ANOVA despite the experimental design requiring a two-way ANOVA. Their justification that this approach follows multiple references is not a scientific argument. Additionally, their claim that more experiments would be necessary to perform a two-way ANOVA is incorrect. The two-way ANOVA can be run using n=10.  The Authors acknowledge that they will consider my suggestion in future studies, but this does not correct the flaw observed here. However, using two-way ANOVA in this study would allow them to analyze possible interaction between dexamethasone and the AJHW treatment and determine which AJHW concentration of can counteract the dexamethasone effect. Finally, the Authors should include in the text the F values of the two-way ANOVA.

Author Response

Reviewer 2 (R2)

Comments 1: 
One is that the text does not explicitly mention that the CSA was analyzed using the minimal Feret’s diameter. This detail is mentioned in the rebuttal letter but not in the manuscript. Since the Authors changed the data representation from bars plot to box-plot as requested by the Reviewer, it is unclear whether they actually recalculated CSA using the minimal Feret's diameter technique or simply changed the data representation.

Response 1:
Thank you for your valuable feedback. We acknowledge the issue and will revise the manuscript to include this detail. Additionally, we will update the Materials and Methods section to clearly include this information.

We appreciate your insightful comment and your effort in improving the clarity and accuracy of our study.

Comments 2: 
The second is that the Authors continue to assume that the antioxidant enzymes activity values found in the serum reflect those in muscle tissue. However, without proper experimental validation, this remains mere speculation. The mentioned enzymes can also be found in erythrocytes, which have high expression of these enzymes, or from other than muscle tissues. The Authors should clarify this in the discussion and acknowledge the potential sources of these enzymes.

Response 2:
We appreciate your insightful comment on this matter.
As per your advice, we will revise the manuscript to clearly describe this limitation

Thank you for your advice.

Comments 3: 

The third point is, again, the statistical analysis. The Authors persist in using the one-way ANOVA despite the experimental design requiring a two-way ANOVA. Their justification that this approach follows multiple references is not a scientific argument. Additionally, their claim that more experiments would be necessary to perform a two-way ANOVA is incorrect. The two-way ANOVA can be run using n=10.  The Authors acknowledge that they will consider my suggestion in future studies, but this does not correct the flaw observed here. However, using two-way ANOVA in this study would allow them to analyze possible interaction between dexamethasone and the AJHW treatment and determine which AJHW concentration of can counteract the dexamethasone effect. Finally, the Authors should include in the text the F values of the two-way ANOVA.

Response 3:
Thank you for your valuable feedback.

In this experiment, we first compared the changes between the untreated control group (G1) and the dexamethasone-treated group (G2), which was induced for muscle loss. Following this, we compared G2 with groups G3, G4, G5, and G6, which were treated with different concentrations of AJHW in addition to dexamethasone.

This approach was designed to assess potential differences between G2 and the AJHW-treated groups (G3, G4, G5, and G6). Based on these experimental conditions, we conducted a one-way ANOVA analysis.

To ensure the validity of our experimental design and statistical analysis, we consulted SPSS statistics expert Mal Kum Han [1]. According to the expert, given the objectives of our experiment and the data obtained, a two-way ANOVA analysis was not applicable. Additionally, under the current conditions, the expert confirmed that a one-way ANOVA along with a t-test is the most appropriate statistical approach.

We have made every effort to incorporate your suggestions as much as possible. However, we repeat that applying a two-way ANOVA in this study would not be suitable for ensuring the statistical significance of our experimental results

Additionally, in our research team's three recently published studies on muscle atrophy, we applied a similar experimental design and performed one-way ANOVA analysis [2,3,4]. During the review and revision processes of these studies, no concerns were raised regarding the use of one-way ANOVA. Based on these past cases, we believe that one-way ANOVA remains a valid statistical approach for the current study. Furthermore, the conclusions from our expert consultant further support this decision.

We appreciate your understanding regarding this matter.

  1. Han Mal Kum, Ph.D., CEO of Research and LAB, Professor at Hanyang Cyber University. Earned a Ph.D. in Advertising and Public Relations from Hongik University and authored two books on SPSS statistical analysis (ISBN: 978-89-5566-252-8 , 979-11-86821-74-9). Currently conducting research in data analysis and research methodology and provided consultation for this study.

  2. Kim, M.S.; Park, S.; Kwon, Y.; Kim, T.; Lee, C.H.; Jang, H.; Kim, E.J.; Jung, J.I.; Min, S.; Park, K.-H.; et al. Effects of Ulmus macrocarpa Extract and Catechin 7-O-β-D-apiofuranoside on Muscle Loss and Muscle Atrophy in C2C12 Murine Skeletal Muscle Cells. Current Issues in Molecular Biology 2024, 46, 8320-8339, doi:https://doi.org/10.3390/cimb46080491.

  3. Lee, C.H.; Kwon, Y.; Park, S.; Kim, T.; Kim, M.S.; Kim, E.J.; Jung, J.I.; Min, S.; Park, K.-H.; Jeong, J.H.; et al. The Impact of Ulmus macrocarpa Extracts on a Model of Sarcopenia-Induced C57BL/6 Mice. International Journal of Molecular Sciences 2024, 25, doi:10.3390/ijms25116197.

  4. An, D.H.; Lee, C.H.; Kwon, Y.; Kim, T.H.; Kim, E.J.; Jung, J.I.; Min, S.; Cheong, E.J.; Kim, S.; Kim, H.K.; et al. Effects of Alnus japonica Hot Water Extract and Oregonin on Muscle Loss and Muscle Atrophy in C2C12 Murine Skeletal Muscle Cells. Pharmaceuticals 2024, 17, doi:10.3390/ph17121661.

Round 3

Reviewer 2 Report (New Reviewer)

Comments and Suggestions for Authors

The Authors addressed most of the suggestions. 

Author Response

Thank you for your kind comments. 

This manuscript is a resubmission of an earlier submission. The following is a list of the peer review reports and author responses from that submission.

Round 1

Reviewer 1 Report

Comments and Suggestions for Authors

The authors and their colleagues investigated the beneficial effects of natural compounds against sarcopenia, identifying that a hot water extract of Alnus japonica exhibited positive effects. Specifically, the study evaluated its impact on dexamethasone-induced skeletal muscle weight loss and motor function decline. However, some results require further clarification, as outlined in the following questions:

1. Could the authors explain the difference in retention times of Hirsutanonol and Hirsunenone in Figures 2 and 3? Are these peaks truly indicative of the same molecule? Additionally, please provide the mass spectra corresponding to Figures 2C and 2D.

2. Although the study mentions MS/MS measurements, the m/z values of the precursor ion and fragment ion are not explicitly listed. Please include these values for clarity.

3. Regarding the treadmill exercise test, could the authors clarify its purpose? Was it intended to measure endurance exercise capacity or muscle exertion tension? Moreover, given the considerable individual variation typically observed in treadmill tests, the reported data shows an unusually small standard error. It would be helpful to include results in a format that displays individual data points for greater transparency.

4. Could the authors provide the rationale for measuring FoxO3a instead of FoxO1, particularly since FoxO1 is a primary molecule involved in insulin signaling?

5. The Western blot bands in Figure 7.8, particularly for mTOR and Bcl2, appear indistinct. For a more comprehensive evaluation of protein synthesis, it would be appropriate to also include the phosphorylation status of 4EBP and S6K. Please provide these results if available.

6. The results for Group 3 (AJHW 20 mg/kg group) indicate significant improvement in muscle atrophy despite no observed improvement in Atrogin1 and Murf1 gene expression. Could the authors elaborate on this finding?

Author Response

Reviewer 1

Comments 1: Could the authors explain the difference in retention times of Hirsutanonol and Hirsutnenone in Figures 2 and 3? Are these peaks truly indicative of the same molecule? Additionally, please provide the mass spectra corresponding to Figures 2C and 2D.

Response 1:
Thank you for your feedback. The difference in retention times observed in Figures 2 and 3 is due to the difference in analytical instruments used. For the analysis in Figure 2, we used the Waters 2695 HPLC System and the Waters 2487 HPLC UV/Vis Detector, whereas for Figure 3, the AB SCIEX System was applied, resulting in the variation in retention times. Additionally, previous studies have also identified this difference in retention time [1-5]. However, despite using entirely different instruments, the retention time difference is approximately 1 minute, and the patterns of the analysis results are similar, which we believe is well within an acceptable range. Additionally, we have updated the Figure 2 to include the mass values as you suggested.

  1. Choi, S.E. Chemotaxonomic Significance of Oregonin in Alnus Species. Asian Journal of Chemistry 2013, 25, 6989-6990, doi:10.14233/ajchem.2013.15090.
  2. Jang, H.; Park, S.; Kim Seong, G.; Bae Seung, B.; Min Hee, J.; Lee Chan, O.; Kim Hee, K.; Kim, J.-K.; Choi Sun, E. Chemotaxonomic Significance of Oregonin in Alnus japonica Native to Baekdudaegan Mountain Range in Korea. Journal of Forest and Environmental Science 2024, 40, 53-63, doi:10.7747/JFES.2024.40.1.53.
  3. Lee, C.H.; Kwon, Y.E.; Kim, S.S.; Kim, H.J.; Kim, H.K.; Kim, J.-K.; Cheong, E.J.; Choi, S.E. Chemotaxonomic significance of catechin 7-O-beta-D-apiofuranoside in Ulmus species native to Asia. Forest Science and Technology 2024, 20, 249-257, doi:10.1080/21580103.2024.2354267.
  4. Lee, C.H.; Kwon, Y.; Park, S.; Kim, T.; Kim, M.S.; Kim, E.J.; Jung, J.I.; Min, S.; Park, K.-H.; Jeong, J.H.; et al. The Impact of Ulmus macrocarpa Extracts on a Model of Sarcopenia-Induced C57BL/6 Mice. International Journal of Molecular Sciences 2024, 25, doi:10.3390/ijms25116197.
  5. An, D.H.; Lee, C.H.; Kwon, Y.; Kim, T.H.; Kim, E.J.; Jung, J.I.; Min, S.; Cheong, E.J.; Kim, S.; Kim, H.K.; et al. Effects of Alnus japonica Hot Water Extract and Oregonin on Muscle Loss and Muscle Atrophy in C2C12 Murine Skeletal Muscle Cells. Pharmaceuticals 2024, 17, doi:10.3390/ph17121661.

Comments 2: Although the study mentions MS/MS measurements, the m/z values of the precursor ion and fragment ion are not explicitly listed. Please include these values for clarity.

Response 2:
In our study, the structural identification of Hirsutanonol and Hirsutenone was primarily conducted using NMR, while LC-MS/MS was performed to confirm the molecular weight. This approach has been consistently applied in our other studies as well [1-5]. Therefore, we kindly ask for your understanding that, despite the value of your inquiry, we are unable to provide the requested information.

  1. Choi, S.E. Chemotaxonomic Significance of Oregonin in Alnus Species. Asian Journal of Chemistry 2013, 25, 6989-6990, doi:10.14233/ajchem.2013.15090.
  2. Jang, H.; Park, S.; Kim Seong, G.; Bae Seung, B.; Min Hee, J.; Lee Chan, O.; Kim Hee, K.; Kim, J.-K.; Choi Sun, E. Chemotaxonomic Significance of Oregonin in Alnus japonica Native to Baekdudaegan Mountain Range in Korea. Journal of Forest and Environmental Science 2024, 40, 53-63, doi:10.7747/JFES.2024.40.1.53.
  3. Lee, C.H.; Kwon, Y.E.; Kim, S.S.; Kim, H.J.; Kim, H.K.; Kim, J.-K.; Cheong, E.J.; Choi, S.E. Chemotaxonomic significance of catechin 7-O-beta-D-apiofuranoside in Ulmus species native to Asia. Forest Science and Technology 2024, 20, 249-257, doi:10.1080/21580103.2024.2354267.
  4. Lee, C.H.; Kwon, Y.; Park, S.; Kim, T.; Kim, M.S.; Kim, E.J.; Jung, J.I.; Min, S.; Park, K.-H.; Jeong, J.H.; et al. The Impact of Ulmus macrocarpa Extracts on a Model of Sarcopenia-Induced C57BL/6 Mice. International Journal of Molecular Sciences 2024, 25, doi:10.3390/ijms25116197.
  5. An, D.H.; Lee, C.H.; Kwon, Y.; Kim, T.H.; Kim, E.J.; Jung, J.I.; Min, S.; Cheong, E.J.; Kim, S.; Kim, H.K.; et al. Effects of Alnus japonica Hot Water Extract and Oregonin on Muscle Loss and Muscle Atrophy in C2C12 Murine Skeletal Muscle Cells. Pharmaceuticals 2024, 17, doi:10.3390/ph17121661.

Comments 3: Regarding the treadmill exercise test, could the authors clarify its purpose? Was it intended to measure endurance exercise capacity or muscle exertion tension? Moreover, given the considerable individual variation typically observed in treadmill tests, the reported data shows an unusually small standard error. It would be helpful to include results in a format that displays individual data points for greater transparency.

Response 3:
The treadmill exercise test was conducted to evaluate exhaustion time. Exhaustion time was directly measured as an outcome of the experiment, and exercise capacity was calculated based on the formula provided in the manuscript. Formula:

Exercise capacity (J, kg*m2*s-2) =

body weight (kg) × speed (m/s) × time (s) × grade × 9.8 m/s2

(1)

In response to your insightful question, we would like to provide the specific values for both results as follows. Additionally, the manuscript includes figures based on supplementary table data.

 Supplementary Table 1. Exhaustion time (min) and Exercise capacity (J)

n

G1

G2

G3

G4

G5

G6

Exhaustion time

(min)

1

45.9

17.5

20.2

20.3

30.7

22.3

2

56.2

20.5

22.0

23.3

34.3

21.2

3

54.7

19.7

24.6

29.1

28.5

23.2

4

47.2

23.9

18.6

26.7

24.3

25.1

5

41.5

22.0

19.3

30.4

25.1

23.7

6

25.8

17.6

23.4

23.8

34.7

19.4

7

41.4

24.9

24.5

18.9

25.1

21.2

8

51.0

22.5

18.7

21.3

23.6

30.1

9

28.9

18.4

18.1

26.4

24.6

29.6

10

31.3

20.7

26.3

28.1

25.2

25.3

Exercise capacity

(J)

1

2836.7

792.8

920.9

924.3

1755.0

1007.4

2

3656.5

1006.5

932.6

1110.2

1618.0

891.8

3

3243.7

908.7

1191.0

1538.6

1639.1

1007.2

4

3232.1

1088.6

756.5

1279.6

1090.8

1086.7

5

2760.3

1063.4

833.4

1610.5

1130.1

1058.7

6

1416.3

720.9

1166.8

1055.5

1762.9

743.2

7

2623.7

1150.8

1152.1

992.4

1319.5

1045.2

8

3041.6

1068.2

714.9

882.0

1107.6

1579.5

9

1550.6

750.3

726.7

1250.0

1131.5

1530.0

10

2149.6

903.6

1397.3

1413.8

1352.8

1203.4

Supplementary Table 2. Average, Standard Deviation of Exhaustion time (min) and Exercise capacity (J)

G1

G2

G3

G4

G5

G6

Exhaustion time

(min)

Average

42.4

20.8***

21.6

24.8

27.6

24.1

Standard Deviation

3.4

0.8

0.9

1.2#

1.3##

1.1#

Exercise capacity

(J)

Average

2651.1

945.4***

979.2

1205.7#

1390.8###

1115.3

Standard Deviation

233.0

48.3

74.1

80.7

88.0

82.7

Values are expressed as the mean ± SEM.
*** p < 0.001 significantly different from that of G1 group. (G2).

# p < 0.05, ## p < 0.01, ### p < 0.001 significantly different from that of G2 group. (G3, G4, G5, G6).

Comments 4: Could the authors provide the rationale for measuring FoxO3a instead of FoxO1, particularly since FoxO1 is a primary molecule involved in insulin signaling?

 Response 4: We agree with your opinion that FoxO1 is a representative factor in insulin signaling, and we appreciate your comment. However, the FoxO3a factor is also highly relevant factor, and this has been confirmed through continued research [1,2,3], which is why we continued to examine FoxO3a in this study. We agree with your assessment that FoxO1 is a representative factor in insulin signaling, and we appreciate your comment.

  1. Kim, M.S.; Park, S.; Kwon, Y.; Kim, T.; Lee, C.H.; Jang, H.; Kim, E.J.; Jung, J.I.; Min, S.; Park, K.-H.; et al. Effects of Ulmus macrocarpa Extract and Catechin 7-O-β-D-apiofuranoside on Muscle Loss and Muscle Atrophy in C2C12 Murine Skeletal Muscle Cells. Current Issues in Molecular Biology 2024, 46, 8320-8339, doi:https://doi.org/10.3390/cimb46080491.
  2. Lee, C.H.; Kwon, Y.; Park, S.; Kim, T.; Kim, M.S.; Kim, E.J.; Jung, J.I.; Min, S.; Park, K.-H.; Jeong, J.H.; et al. The Impact of Ulmus macrocarpa Extracts on a Model of Sarcopenia-Induced C57BL/6 Mice. International Journal of Molecular Sciences 2024, 25, doi:10.3390/ijms25116197.
  3. An, D.H.; Lee, C.H.; Kwon, Y.; Kim, T.H.; Kim, E.J.; Jung, J.I.; Min, S.; Cheong, E.J.; Kim, S.; Kim, H.K.; et al. Effects of Alnus japonica Hot Water Extract and Oregonin on Muscle Loss and Muscle Atrophy in C2C12 Murine Skeletal Muscle Cells. Pharmaceuticals 2024, 17, doi:10.3390/ph17121661.

Comments 5: The Western blot bands in Figure 7.8, particularly for mTOR and Bcl2, appear indistinct. For a more comprehensive evaluation of protein synthesis, it would be appropriate to also include the phosphorylation status of 4EBP and S6K. Please provide these results if available.

 Response 5:
While we would have liked to include data on the mTOR and Bcl2-related factors 4EBP and S6K, as you kindly suggested, the timing of the revision step unfortunately limits our ability to conduct additional research at this stage. As a result, we are unable to provide the requested data on 4EBP and S6K at this time.

However, we greatly appreciate your thoughtful suggestion, which will certainly contribute to the enhancement of our research. Although the in vivo study has already been completed, and it is physically impossible to generate the required data in time for this revision, we do plan to explore experiments related to 4EBP and S6K in our future studies.

Comments 6: The results for Group 3 (AJHW 20 mg/kg group) indicate significant improvement in muscle atrophy despite no observed improvement in Atrogin1 and Murf1 gene expression. Could the authors elaborate on this finding?

 Response 6:
The study results for the Atrogin1 and MuRF1 factors in Group 3 did not show statistical significance, although a trend toward decrease was observed. This appears to be attributed to the complex interactions not only among these factors (Atrogin1, MuRF1) but also with others. Our research team will consider your valuable comments and design a more detailed and in-depth study for the low-dose treatment group in future research. Thank you for your insightful feedback.

Reviewer 2 Report

Comments and Suggestions for Authors

In this pilot study, the effects of warm water extract of Alnus japonica on sarcopenia and muscle atrophy were evaluated. Biological activity studies focusing on oxidative stress, apoptosis and muscle synthesis and degradation, which influence sarcopenia, were performed.

The study is very complex, logically organized, meets the criteria for organizing a study. I congratulate the work team for realizing this study.

A few suggestions:

ü Specify the number of animals in each group,

ü The information in the conclusions chapter (lines 856-900) should be included in the discussion chapter

ü Systematization of conclusions

Author Response

Reviewer 2

Comments 1: Specify the number of animals in each group,

Response 1: Thank you for your feedback. A total of ten animals were utilized in each group. I will ensure that this information is included in the manuscript.

Comments 2: The information in the conclusions chapter (lines 856-900) should be included in the discussion chapter

Response 2: Thank you for your valuable review. We will relocate several parts to the Discussion section.

Comments 3: Systematization of conclusions

Response 3: After relocating specific sections to the Discussion part, we reorganized the content in the Conclusion. Thank you for your feedback on this matter.

Round 2

Reviewer 1 Report

Comments and Suggestions for Authors

Thank you for providing detailed explanations in response to my review comments. I have no further comments at this time.